



# Grassland yield estimations - potentials and limitations of remote sensing, process-based modelling and field measurements

Sophie Reinermann[1], Carolin Boos[2], Andrea Kaim[3,4], Anne Schucknecht[5], Sarah Asam[6], Ursula Gessner[6], Sylvia H. Annuth[3], Thomas M. Schmitt[2,3], Thomas Koellner[3], Ralf Kiese[2]

[1]University of Würzburg, Institute of Geography and Geology, Department of Remote Sensing, Am Hubland, 97074 Würzburg, Germany
[2]Karlsruhe Institute of Technology (KIT), Institute of Meteorology and Climate Research, Atmospheric Environmental Research (IMKIFU), Garmisch-Partenkirchen, Germany
[3]University of Bayreuth, Professorship of Ecological Services, Bayreuth Center of Ecology and Environmental Research (BayCEER), Universitaetsstr. 30, 95447 Bayreuth, Germany
[4]Helmholtz Centre for Environmental Research - UFZ, Department of Computational Landscape Ecology, Leipzig, Germany
[5]OHB System AG, Image Simulation and Processing Team, Manfred-Fuchs-Str. 1, 82234 Wessling, Germany
[6]German Aerospace Center (DLR), German Remote Sensing Data Center (DFD), Oberpfaffenhofen, Wessling 82234,
Germany

*Correspondence to*: Sophie Reinermann (sophie.reinermann@dlr.de)

**Abstract.** Grasslands make up the majority of agricultural land and provide fodder for livestock. Information on grassland yield is very limited as the fodder is directly used at the farms. Data on grassland yields would be needed, however, to inform politics and stakeholders on grassland ecosystem services and inter-annual variations. Grassland yield patterns are

often varying on small scales in Germany and estimations are further complicated by missing information on grassland management. Here, we present three different approaches to estimate annual grassland yield for a study region in southern Germany. We apply (i) a model derived from field samples, satellite data and mowing information (RS), (ii) the biogeochemical process-based model LandscapeDNDC (LDNDC) and (iii) a rule-set approach based on field measurements and spatial information on grassland productivity (RVA) to derive grassland yields per parcel for the Ammer catchment area

in 2019. All three approaches reach plausible results of annual yields of around 4-9 t/ha and show overlapping as well as diverging spatial patterns. For example, direct comparisons show that higher yields were derived with LDNDC compared to RS and RVA, in particular related to the first cut and for grasslands mown only one or two times per year. The mowing frequency was found to be the most important influencing factor for grassland yields of all three approaches. There were no significant differences found in the effect of abiotic influencing factors, such as climate or elevation, on grassland yields

derived from the different approaches. The potentials and limitations of the three approaches are analysed and discussed in depth, such as the level of detail of required input data, or the capability of regional and inter-annual yield estimations. For the first time, three different approaches to estimate grassland yields were compared in depth resulting in new insights in their potentials and limitations. Grassland productivity maps provide the basis for long-term analyses of climate and management impacts and comprehensive studies of the functions of grassland ecosystems.




## 1 Introduction

Grassland ecosystems provide fodder for livestock, apart from many other ecosystem services, such as carbon storage, provision of habitats, water purification, recreation and erosion control (Bengtsson et al., 2019; Le Clec'h et al., 2019; Gibon, 2005; Gibson, 2009; Richter et al., 2021; White et al., 2000). In Germany, grasslands cover almost one third of the

agriculturally used area (Statistisches Bundesamt, 2023) and are of central importance for the meat and dairy industry (Schoof et al., 2020b; Soussana and Lüscher, 2007). In large parts of Europe, grassland ecosystems are managed, hence strongly shaped by human activities. In Germany, for example, almost all of the grassland is under some form of agricultural use; i.e. grazed and/or mown in different frequency (Dengler et al., 2014; Schoof et al., 2020a, c). Grassland management and use intensity, i.e. the number and timing of grazing and/or mowing as well as fertilization events, have a strong impact

on grassland functions and ecology (Gossner et al., 2016; Neyret et al., 2021; Socher et al., 2012). Apart from climate and soil conditions, grassland management determines the productivity, thus yields, and species diversity of these ecosystems (Gilhaus et al., 2017). In Germany, grasslands are managed on small units (parcels) individually, resulting in a wide variety of combinations of the number and timing of mowing events on small spatial scales. As a consequence, grassland landscapes can show high spatial and temporal variability in their biomass availability and species composition (Gerowitt et al., 2013).


Grassland biomass is usually directly used on farms as fodder for livestock and not traded, which is why there is usually no data on grassland yields resulting from sales statistics. The lack of information on yields exacerbates extensive spatio-temporal analyses of drivers of grassland productivity, as well as modelling of grassland ecosystem services, e.g. nitrogen and carbon fluxes. Long-term effects of climate change as well as short-term weather extremes influence grassland

productivity and yields (Beniston, 2003; Berauer et al., 2019). In the Alpine and pre-Alpine regions of southern Germany this is of particular importance, since temperature increases twice as fast as the global average (Auer et al., 2007; Kiese et al., 2018). In addition, drought and heat episodes are expected to increase in the region. Therefore, information on grassland yields and the dependency on climate conditions is needed to support the planning of fodder production and imports for farmers and to inform administration and politics. Furthermore, information on grassland yields is required for a

comprehensive assessment of grassland ecosystem services and sustainable management also under changing climate conditions. Despite these information needs, continuous and large-scale monitoring is lacking.

There are different approaches to retrieve grassland yield information. Ground-measurements alone such as cutting and removing herbage from the grassland for direct analysis or estimating yields by the use of a rising plate meter are usually

time intensive, can hardly provide regular information and might not represent the conditions on broader spatial scales (Murphy et al., 2021). This holds in particular in grassland ecosystems characterized by a high small-scale variability, like in southern Germany. To retrieve spatially continuous and multi-temporal information, grassland yields can be (i) modelled empirically in different degree of complexity, e.g. taking in-situ and remote sensing data into account, (ii) modelled bio-





geochemically, e.g. with process-based models, or (iii) derived from simple rule sets used by authorities based on yield
surveys and further spatially extensive data, e.g. elevation and soil fertility index.

Remotely sensed reflectance, and in particular vegetation indices derived from them, depict vegetation greenness, structure
and photosynthetic activity and, thus, relate to vegetation biomass (Holtgrave et al., 2020; Huete et al., 2002). Grassland
traits, such as above-ground biomass, can be estimated using an empirical relationship employing remote sensing and in-situ
data to train and validate models, as shown in many studies summarized in Reinermann et. al. (2020). Space-borne remote
sensing-based biomass models have been applied in many different grassland ecosystems using various sensors and
regression models. Using satellite remote sensing data to quantify vegetation properties enables large-scale, continuous,
reproducible and comparatively cost-sensitive monitoring. Compared with the relatively frequent application of empirical
remote sensing data-based biomass models for mostly grazed grassland ecosystems (Wu et al., 2024; Yao and Ren, 2024),
the number of studies using this approach for grasslands dominated by mowing is more limited (Reinermann et al., 2020).
Previous studies from regions characterized by mown grasslands investigated the potential of various vegetation indices
derived from medium resolution sensors (moderate resolution imaging spectroradiometer (MODIS), moderate resolution
imaging spectrometer (MERIS)) to estimate grassland biomass for single sites in Ireland and the Netherlands (Ali et al.,
2017a; Ullah et al., 2012). Based on Landsat and Sentinel-2, grassland biomass and height were estimated for study regions
in Germany, France, Spain and Austria using various regressors, such as multi-linear regression, random forest or deep
learning models (Barrachina et al., 2015; Dusseux et al., 2022; Eder et al., 2023; Muro et al., 2022; Schwieder et al., 2020).
However, despite the strong influence of grassland management on the productivity, to our knowledge none of the previous
remote sensing-based studies have directly included mowing information in the biomass estimation approach. Further,
grassland biomass estimates are only a snapshot in time. In particular for grasslands dominated by frequent mowing
activities, the amount of standing biomass varies a lot in the course of a year. A single biomass estimation is therefore not
sufficient to inform on annual grassland productivity and yields. One way to approach this is the combination of multi-
temporal biomass estimations informed by timing of mowing events to retrieve annual grassland yields. To our knowledge,
there is no remote sensing-based study that estimated annual grassland yields using this approach so far.

Another method to obtain grassland yield estimates are deterministic process-based models, like LandscapeDNDC (Haas et
al., 2013; Kraus et al., 2015; Petersen et al., 2021), Daycent (Del Grosso and Parton, 2019; Parton et al., 1998), PaSIM
(Riedo et al., 1998, 2000), LPJmL (Bondeau et al., 2007; Schaphoff et al., 2018), APSIM (Holzworth et al., 2014), or
ORCHIDEE-GM (Chang et al., 2013). The general idea is to describe the most relevant processes determining plants or
plant community behavior and their dependence on environmental conditions by a set of differential equations connecting
atmospheric, plant and soil processes. An advantage of process-based models is the possibility to assess all spatial levels
ranging from the site (Chang et al., 2013; Liebermann et al., 2020; Petersen et al., 2021) to continental (Vuichard et al.,
2007) and global (Rolinski et al., 2018) scale. Additionally, the application of process-based models opens the possibility to




evaluate ecosystem productivity under various scenarios including climate change (Petersen et al., 2021), or management changes like adaptions in fertilization regimes (Hong et al., 2023; Reis Martins et al., 2024), or shifts in cutting frequencies

(Rolinski et al., 2018). While model input data is generally available for model development and testing at the site scale, up-scaling of results is often limited by uncertainty or even lack of detailed information particularly on grassland management.

A third approach to estimate grassland yield is by making use of measurements from field experiments or regional census statistics (Smit et al., 2008). In Germany, some federal states provide reference values for grassland yields at county level

that can be used by farmers to derive their grassland's fertilizer requirements. For instance, the reference values provided in the guideline for fertilization of crop- and grassland by the Bavarian State Institute for Agriculture (LfL) (LfL, 2018) are aggregated values for Bavaria based on LfL internal research and field experiments (Diepolder et al., 2016). Thus, grassland yields can be derived from rule-based calculations based on grassland yield reference values and data e.g. on soil properties, climate, and use intensity to adapt these values to local conditions and management.


Here, three approaches to derive grassland yield – either established in the scientific community or used by authorities, however optimized for the study region – are applied for the same region and year. Annual yields are compared and advantages and disadvantages are highlighted. We estimate annual grassland yields for a study area in southern Germany in 2019 using (a) a novel empirical satellite remote sensing model (RS), which is compared with results of (b) a process-based

biogeochemical model (LandscapeDNDC) and with (c) a simple rule-set reference value approach used by authorities (RVA). To represent grassland management intensity, all three approaches use information on satellite-derived mowing dates retrieved from Reinermann et al. (2023) and (2022). To examine under which conditions the results of the three different approaches differ, we examined the influence of various factors – management and climate – on the grassland yields resulting from the three methods. By examining the spatial and temporal patterns of yields derived from the three

approaches and analyzing the influence of various factors, we assess the differences and similarities in the methods. We aim to determine which method best represents specific conditions and under which circumstances it is most reliable.

## 2 Study area

The study area is located in the pre-Alpine and Alpine region of southern Germany (Figure 1) and consists of the broader Ammer catchment area including the Pre-Alpine TERENO Observatory (Kiese et al., 2018). The area belongs to the

temperate oceanic climate according to Köppen and Geiger (Kottek et al., 2006). The mean annual temperature was 8.9 °C in 2019, the long-term average (2012-2021) is 8.1 °C for the region. The mean precipitation sum was 1175 mm in 2019 and the long-term average 1141 mm (Boos et al., 2024; Petersen et al., 2021). The elevation of the study area with grassland land use ranges between 500 and 1100 m a.s.l. with grasslands dominating agricultural land use which totals to about 38% of the region area (Kiese et al., 2018). In the region, grasslands are of economic importance, in particular for meat and dairy



production, but also for tourism (Schmitt et al., 2024; Soussana and Lüscher, 2007). The grasslands of the Ammer region are grazed and/or mown at intensities ranging from extensive (with one to two mowing events) to highly intensive use with up to six mowing events per year (Reinermann et al., 2022, 2023). The timing of the management activities varies from grassland parcel to parcel. Here, we focus on meadows and mowing pastures which make up around 657 km² (27138 parcels).

**Figure 1: Study area in southern Germany showing land use, elevation and hexagon averages of 2019 of the mowing frequency, mean annual air temperature, mean annual precipitation, and elevation (see Section 3.2.2). The hexagon diagonal size is 1 km.**



## 3 Material and methods

### 3.1 Spatial and field data

#### 3.1.1 Spatial management data

All three yield modelling approaches used the same parcel boundary and mowing information data. Parcel boundaries were taken from the EU's Integrated Administration and Control System (IACS) provided by the Bavarian State Institute for Agriculture (LfL). The same data source was used to exclude all parcels which were not used as meadows or mowing pastures in 2019.

The dates of mowing events originate from Reinermann et al. (2023) and (2022) where mowing events were detected based on Sentinel-2 time series. However, differing from the original approach, for this study no grassland mask was used to ensure that all meadows and mowing pastures identified by the IACS data are covered. To transfer the 10x10 m² pixel-based mowing dates to parcel level, detected dates within a time frame of three weeks were agglomerated per parcel using the majority vote. Only when at least for 20 % of the parcel the date was detected, the mowing event remained in the dataset.

Regional validation using information from farmers and webcam images showed an accuracy (F1-Score) of 0.65 for the mowing dates on parcel level in the Ammer region in 2019.

#### 3.1.2 Biomass field data

To train and validate the empirical remote sensing model as well as for regional quality evaluation of LandscapeDNDC, in-

situ biomass measurements were used. A total of 14 grassland plots (20 m x 20 m) in the Ammer region were sampled in the period 2019-2021 to obtain in-situ above-ground biomass (AGB) information (for sampling design see Schucknecht et al. (2023) and Schucknecht et al. (2020)). The sampling plots are constituted by homogeneous vegetation coverage and placed to be representative for the entire grassland parcel. The sampled grassland parcels are characterized by different land management intensities ranging from one to six mowing events per year. The sampling campaigns took place at multiple

times during the growing season to ensure that biomass samples from a variety of growth stages before and after mowing events were included. For each plot, above-ground biomass was collected on four randomly placed subplots of 50 cm x 50 cm. To account for mowing height, AGB was sampled from >7cm on all subplots and complemented by one biomass sampling from 2-7 cm vegetation height at one of the four subplots. The samples were dried for at least 48 hours until constant weight at 60 °C and weighed. The weight of the dried biomass of the four samples from above 7 cm were averaged

and added to the measurements of 2-7 cm to obtain total AGB per site and date as well as scaled to 1 m². In total, 111 biomass samples were collected.



## 3.2 Yield estimation approaches

The three grassland yield estimation approaches applied in this study, i.e., the remote sensing (RS) approach, the process-
based LandscapeDNDC model approach (LDNDC) and an estimation based on reference values (RVA), are described in
detail in Sect. 3.2.1-3.2.3. (Figure 2).

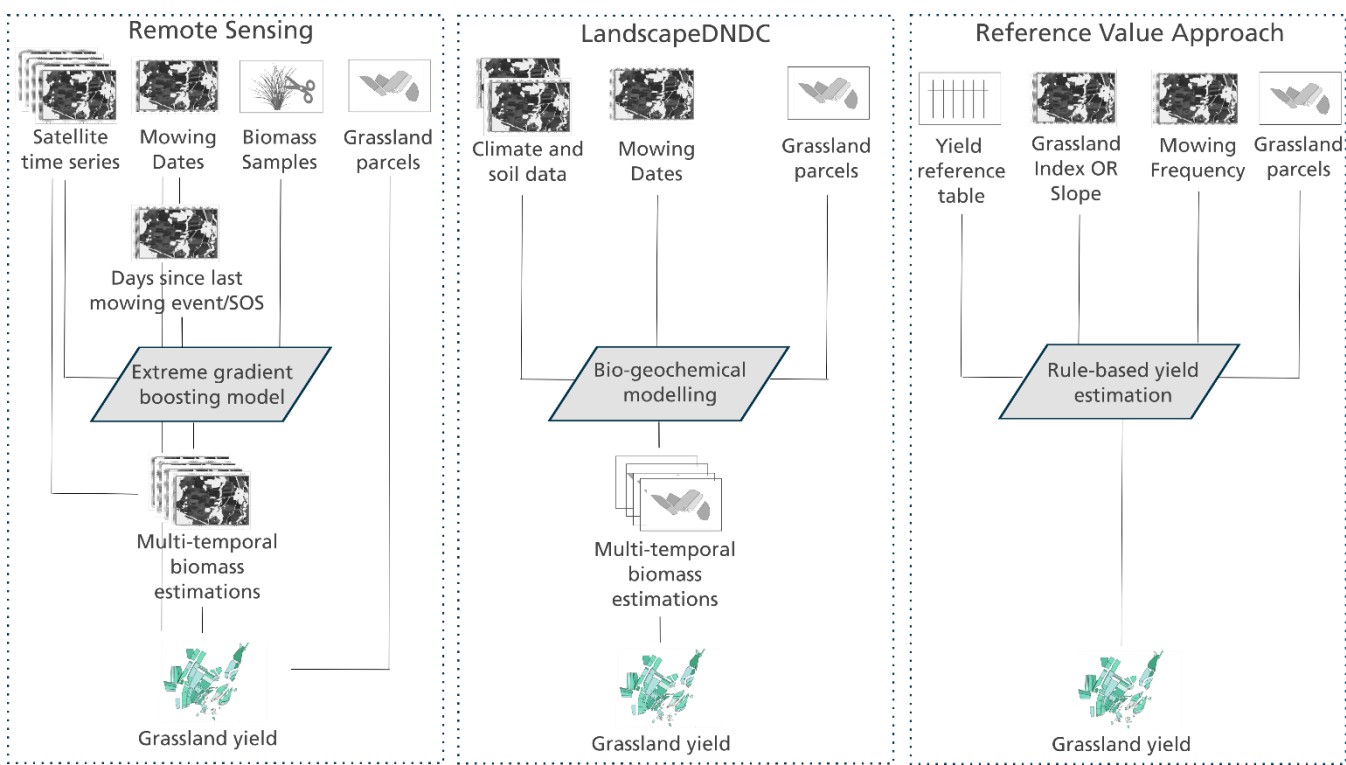

**Figure 2: Conceptual scheme of the three yield estimation approaches.**

### 3.2.1 Remote sensing

For the remote sensing model, MAJA (version 3.3) Sentinel-2 (S2) level 2a (Hagolle et al., 2017) time series from the years
2019-2021 and tiles 32TPT and 32UPU were used to match the biomass sampling campaigns. The optical satellite
reflectance data consists of acquisitions from two identical satellites (S2A and S2B) acquiring information in 12 spectral
bands (Drusch et al., 2012). The bands used here are the 10 bands relevant for vegetation monitoring, namely bands 2, 3, 4,
5, 6, 7, 8, 8A, 11 and 12, covering the red, green, vegetation red edge, near-infrared and shortwave-infrared wavelengths.
The bands which have a 20 m spatial resolution were resampled to 10 m by the nearest-neighbour method to achieve a
consistent spatial resolution of 10 m for all bands.



Based on S2 satellite data and in-situ total AGB samples (Section 3.1.2), an empirical model was trained and optimized to
estimate grassland biomass. Satellite data influenced by clouds, cloud shadows or unfavourable terrain conditions were
excluded according to the MAJA algorithm. The empirical model was built based on the S2 reflectances from the 10 selected
bands and additional spectral indices as predictor variables. Specifically, the Enhanced Vegetation Index (Huete et al., 2002)
(EVI, Equation 1) and the Tasseled Cap Wetness Index (Indexdatabase, 2024; Krauth and Thomas, 1976) (wetness, Equation
2) were calculated and included as they relate to vegetation biomass:

$EVI = \frac{2.5*B8-B4}{B8+6*B4-7.5*B2+1}$,                                                     (1)

Wetness $= 0.1509 * B2 + 0.1973 * B3 + 0.3279 * B4 + 0.3406 * B8 - 0.7112 * B11 - 0.4572 * B12$,          (2)

where B2, B3, B4, B8 and B11 are reflectance bands in the blue, green, red, near infrared and short-wave infrared area,
respectively.


In addition, information on timing of mowing events (Section 3.1.1) was directly included in the modelling process by
adding an additional predictor variable representing the days since the last mowing event. For each S2 acquisition, a layer
was calculated giving the days since the last mowing event on pixel basis. When no mowing event took place before the S2
acquisition, the number of days since the start of the growing season was calculated. To retrieve the start of the growing
season, the Copernicus Land Monitoring Service (CLMS) High Resolution Vegetation Phenology and Productivity (HR
VPP) Start of Season product was used. Further, the S2 acquisition date was included into the mode as predictor variable.
This resulted in 14 input features for the empirical modelling, i.e. 10 spectral bands, 2 spectral indices, the days since last
mowing/start of growing season, and the date of the satellite acquisition.

To prepare the input data for model training, pairs of cloud-free S2 acquisitions and corresponding in-situ biomass samples
were built by allowing a maximum of five days between satellite acquisition and field sampling in both directions. If there
were multiple satellite acquisitions in the allowed range, closer ones were preferred as well as satellite acquisitions after field
sampling dates. It was also checked that there was no mowing event in between a satellite acquisition and a field sampling to
maintain representative data pairs. Due to cloud conditions in 2021, only data from 2019 and 2020 remained in the data table
after this procedure. Data pairs from sampling campaigns from every month between April and October, apart from July and
August, were available.

An extreme gradient boosting model was trained on the input features and the corresponding AGB values (Friedman, 2001).
Initial tests showed that the extreme gradient boosting model outperformed others, such as Random Forest, Support Vector
Machines or multi-linear models. The xgboost package (version 1.5.2) was used in Python. In total, 74 data pairs were
available from which 82% (n=61) were used for training and testing and 18% (n=13) as independent test of the trained



model. A stratified sampling of the test data was conducted to ensure that the value range of the test data was representative. With the data used for training the hyperparameters for an optimized model were searched for, using grid search, 5-fold cross-validation (CV) and ten iterations each. To find the best model the coefficient of determination (R²) was used. The best

model according to the training was then tested against the independent test data set.

The best trained biomass model was applied to estimate the AGB of all available S2 scenes to generate a biomass time series. This biomass time series was used in combination with the mowing dates and IACS parcel information to estimate annual yields per parcel. This was approached by going through the parcel-based mowing dates. For each mowing date, the

pixel-based biomass estimates from all observations of up to three weeks before and one week after the mowing date were extracted. The 95 % percentile was calculated from this biomass data to estimate the yield per mowing event and parcel, minimizing the influence of parcel boundaries. This time frame was used to ensure that the biomass was captured shortly before a mowing event as there is an uncertainty in the timing of the mowing dates. These single mowing event yields were afterwards summed up to annual yields per parcel.

**3.2.2 LandscapeDNDC**

The process-based biogeochemical model LDNDC was run for the whole study region with individual high-quality input data combinations of soil, climate and management for every field (Boos et al., 2024). This became possible by the availability of accurate small-scale grassland soil profile data, interpolated reference climate data based on continuous measurements from weather stations, and cutting dates from remote sensing on a field level.


The model calibration and validation for yields was performed on extensive measurements on lysimeters from the TERENO pre-Alpine observatory covering three sites in different elevations within the study area with intensive and extensive management (Kiese et al., 2018; Petersen et al., 2021). For the harvested dry weight biomass at individual cutting events, coefficients of determination between 0.52 and 0.61 with relative root mean square errors (RRMSEs) (RMSEs) between 0.32

and 0.37 (0.72 t ha$^{-1}$ to 0.92 t ha$^{-1}$) are found for the sites at 864 m a.s.l. and 595 m a.s.l. in 2012-2018 in Petersen et al. (2021). In a further model validation, considering the period 2012-2021 and also adding the mid-elevation site at 769 m a.s.l., yields from individual mowing events were captured with a coefficient of determination (r2) of 0.61 and a RMSE of 0.94 t ha$^{-1}$ (RRMSE 0.39) (Boos et al., 2024). Running the model with regional input data for the sites, which are used for RS-training and validation, compare Sect. 3.1.2., and comparing to standing biomass before cutting events, lead to a

coefficient of determination of 0.67 and a RMSE of 1.46 t ha$^{-1}$ a$^{-1}$ (Boos et al., 2024).

Climate inputs were generated from the reference data of the Climex project (Poschlod et al., 2020; Willkofer et al., 2020), which have been interpolated from station measurements of the German Weather Service into a product of 3-hourly temporal and 500 m x 500 m spatial resolution of virtual climate stations. For this study, we aggregated this data to daily climate





inputs (minimum, maximum, and mean air temperature, precipitation sum, mean relative humidity, mean global radiation, and mean wind speed) and assigned the nearest virtual climate station to every field to run LDNDC. Further, seasonal and yearly mean temperatures and summed precipitations for result analysis were derived. Regional soil data was derived from the soil database of the Bavarian State Office of the Environment (LfU) (LfU, 2020). Only mineral grassland soils were considered and a unique, i.e. with a single profile per polygon, soil map was compiled. More details can be found in Boos et

al. (2024).

The model simulates plant growth depending on factors like photosynthesis, nitrogen and water availability, phenology, and temperature (Petersen et al., 2021). At a prescribed cutting date, the above ground biomass is reduced to a pre-set value for the remaining biomass, which equals the standing biomass after the cutting event and is according to farmers' practice

calibrated to a cutting height of about 7 cm. The harvested biomass from all events in a year is then summed up to calculate annual yields per field. Therefore, the management is another key model driver and was set for every parcel individually. The cutting dates in the study year 2019 were taken from the dataset generated by Reinermann et al. (2022), as described in Sect. 3.1.1. Fertilizer in the form of slurry was applied according to the mowing information following farmers' practice in the study region. For parcels with three or more mowing events per year, the number of manuring events equalled the

number of cuts. For parcels with less than three mowing events per year, the number of fertilizer applications was one less than the number of cuts which corresponds to local farmers practices. The amount of manure varied between 40 and 55 kgNha$^{-1}$ per event and decreased per application. For further details on the applied regional model drivers for LDNDC, see Boos et al. (2024).

For every grassland field simulation (N=27138), climate, soil, and management input were derived from superimposing field boundaries with the respective spatial products. The model (LDNDC revision: 10786, Crabmeat revision: 8136) was run with an hourly time step and the submodels CanopyECM (Grote et al., 2009) as the microclimate module, WatercycleDNDC (Kiese et al., 2011) as the watercycle module, MeTrx (Kraus et al., 2015) as the soil-chemistry module, and PlaMox (Kraus et al., 2016; Liebermann et al., 2020) employing the PhotoFarquhar model (Ball et al., 1987; Farquhar et al., 1980) for

photosynthesis as the physiology module. For a general description of LDNDC and the functioning and interaction of the different sub-modules see Petersen et al. (2021).

### 3.2.3 Reference values approach

The reference values approach (RVA) is mainly based on a look-up table from the Bavarian State Institute for Agriculture (LfL) that includes yield reference values of the farmer's yield for Bavaria for different types of grassland uses and

intensities (number of cutting events, low/medium/high grazing intensity) as well as yield levels (low, medium, high) (see Appendix A1). Apart from the mowing information (Section 3.1.1) cattle numbers (Section 3.1.1) were used to identify the management intensity. We further used the land appraisal dataset (LDBV, 2018) to obtain grassland indices (German



"Grünlandzahl") for each field. This index represents the quality of a location for grassland production, considering factors such as soil type, soil properties, climate, and water availability. The index ranges from 1 (poor) to 100 (best). As the grassland index can vary within a parcel, we assigned the value that covered the largest portion of the parcel area. In cases, where the grassland index was unavailable, we substituted it with the field's maximum slope (ASTER GDEM, 2018) instead.

The yield estimation approach based on reference values is based on Kaim et al. (under review). Table A1 provides reference yield values for meadows and mowing pastures, categorized by different use intensities and yield levels (low, medium, and high). To derive the yield for each grassland parcel, we defined a set of rules based on use intensities and yield levels. We determined the yield of each parcel based on the grassland index, i.e. the higher the grassland index, the higher the yield level. For a few fields, the grassland index was not available due to data gaps and we used the maximum slope to substitute the missing information on grassland site conditions. The assumption was that with increasing slope the management intensity decreases, consequently also the grassland yields. The management intensity dataset described in Sect. 3.1.1 was used to allocate the number of cutting events to each grassland field. For mowing pastures, firstly, the share between mowing and grazing had to be identified in order to determine their use intensities. Mowing pastures with one cutting event were defined as being used up to 60 % for grazing and all mowing pastures with at least two cutting events as being used up to 20 % for grazing. Secondly, the management intensity of mowing pastures was approximated by the farm's stocking rate (SR) for all parcels with zero to three cuts, i.e. the higher the SR, the higher the use intensity. SR is defined as LSU/P, with LSU being the number of cattle per farm in livestock units and P the farm's total grazing area. Mowing pastures with more than three cuts were assumed to have a high use intensity (Table 1). All assumptions regarding slope, share of grazing in mowing pastures, and SR were discussed with and approved by grassland experts from science and the Office of Food, Agriculture and Forestry (AELF) Weilheim, Germany – a local stakeholder from the study region.

**Table 1: Grassland yield level (/grazing intensity) according to grassland index or slope (/stocking rate).**

| Yield level / grazing intensity | Grassland index (GI) | Slope | Stocking rate (SR) |
|---|---|---|---|
| Low | $0 \leq GI \leq 33$ | $50\% \leq slope$ | $0 = SR \leq 1.5$ |
| Medium | $33 < GI \leq 67$ | $25\% \leq slope < 50\%$ | $1.5 < SR \leq 3$ |
| High | $67 < GI \leq 100$ | $0\% \leq slope < 25\%$ | $3 < SR$ |

### 3.3 Spatial aggregation of yield data and comparisons

The annual yields estimated by all three approaches are compared for any individual parcel (N = 27138 meadows and mowing pastures) as well as for mean values of hexagons (1 km diagonal) with a total of 2571 hexagons covering the full



study region. The yields were averaged per hexagon by weighting with the parcel areas. The aggregation on hexagons has several reasons: Firstly, it is not allowed to publish the parcel shapes and locations, due to data privacy regulations, secondly, the visibility of spatial patterns is improved and, lastly, outlier effects are minimized.

To compare the yields per parcel and hexagon resulting from the three different approaches, the Pearson correlation coefficient was calculated. To analyse the effect of influencing factors, the relationships between yields and mowing frequency, temperature, precipitation and elevation were plotted and Pearson correlation coefficients calculated.

## 4 Results

### 4.1 Grassland biomass estimation

The yield estimation methods LDNDC and RS involve an estimation of grassland biomass over time. Time series of AGB for the year 2019 are illustrated in Fig. 3 for three example parcels from the measurement campaign which were used for the RS biomass model training and validation (Section 3.1.2.). The figures include the AGB of in-situ measurements, estimated AGB derived by the RS and LDNDC approach and the annual yield based on all three estimation methods. The temporal pattern of LDNDC biomass estimations shows an increase in spring, drops after mowing and increase thereafter. The

LDNDC biomass at the first cut is the highest. The temporal profile of the RS-based biomass follows the mowing dates less strictly as compared to LDNDC; however, for most mowing events there is a peak before the mowing event, followed by a drop and a regrowth pattern (Figure 3). When comparing the two biomass time series it becomes clear that the yield of the first mowing event from LDNDC is notably higher than from RS. In contrast to that, RS shows only at times a gradual decline in the local maxima during the growing season (e.g. Figure 3 top).





Figure 3: Temporal patterns of grassland AGB estimated by the RS and LDNDC models, in-situ measurements of AGB, annual yields of all three models (on the right of each sub-figure), and mowing dates (horizontal dashed lines) of three grassland parcels in the study area.

The estimated biomasses from RS were validated with a part (18%) of the in-situ measurements that were not used for model training (i.e., the test data). The best RS model – extreme gradient boosting regressor, parameterized with a learning rate of




0.05, a maximum depth of each tree of three and a number of features used in each tree of 40 % – reached an average $R^2$ (CV) of 0.97 and a root mean square error (RMSE) (CV) of 1.84 t ha$^{-1}$ during the internal validation. Band 12 (short-wave infrared), wetness index and days since the last mowing were the most important features according to the relative influence measure. The validation of the model with the test dataset (n=13) lead to a $R^2$ of 0.68 and a RMSE of 1.87 t ha$^{-1}$.


The LDNDC model together with the regional input data was also evaluated against the AGB measurements/in-situ data of the above-mentioned field campaign (Section 2.2.2) in a previous study (Boos et al., 2024). To this end, only biomass measurements from within one week before a mowing event were considered and summed to yearly values per parcel. With this procedure an $R^2$ of 0.67 and a RMSE of 1.46 t ha$^{-1}$ a$^{-1}$ were found.

**4.2 Estimated annual grassland yields**

The spatial patterns of annual grassland yield averaged per hexagon estimated with each method are depicted in Fig. 4. All three models achieve plausible results of annual grassland yields ranging mostly between 3 to 9 t/ha for the Ammer region in 2019. The annual grassland yields of the entire study region are on average 6.5 t ha$^{-1}$ (372.9 kt) estimated by RS, 7.4 t ha$^{-1}$ (445.5 kt) estimated by LDNDC and 6.9 t ha$^{-1}$ (419 kt) estimated by RVA. The yield maps highlight the spatial variability of
grassland yields in the study region, which is consistent in many cases for the three modelling approaches. Noticeable patterns are grasslands with relatively high annual yields in the north of the study region according to the RS and LDNDC models and to a lesser degree also according to the RVA. Grasslands with lower annual yields are present in the northeast and the east of the study area, which is mostly visible in the RS and RVA maps. The centre of the study area shows grasslands with above-average yields mostly based on the RS and even more the LDNDC model results. Grassland in the
south of the study area located within Alpine valleys show lower yields in particular for the RS and RVA maps. The entire south-western part of the region shows overall lower grassland yields from the RS and LDNDC, but not the RVA models (Figure 4).

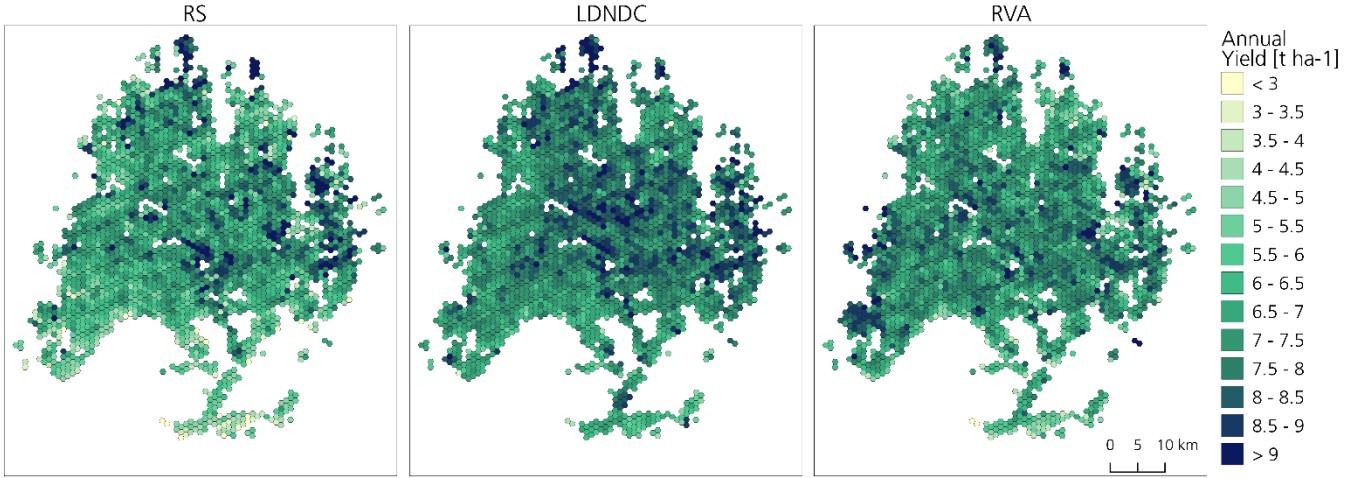



**Figure 4: Spatially aggregated (hexagon diagonal length of 1km) annual yield estimates for meadows and mowing pastures in the**
**study area in 2019 based on Remote Sensing, LandscapeDNDC simulations and the reference values approach. Hexagons for**
**which the grassland area is smaller than 1 ha are not shown.**

The differences in yields based on RS and LDNDC are not concentrated to specific locations but rather distributed throughout the Ammer region (Figure 5). The yield differences between RS and LDNDC to RVA shows in both cases a north-east, south-west pattern as yields derived from RVA are higher in the south-western part of the study region compared

to the yields based on RS and LDNDC. Direct comparisons of hexagon yields reveal that 36 % of the Ammer region shows differences smaller than 1 t ha$^{-1}$ among all three approaches. The standard deviation of yield averages of all three methods shows no distinctive spatial patterns (compare A2).

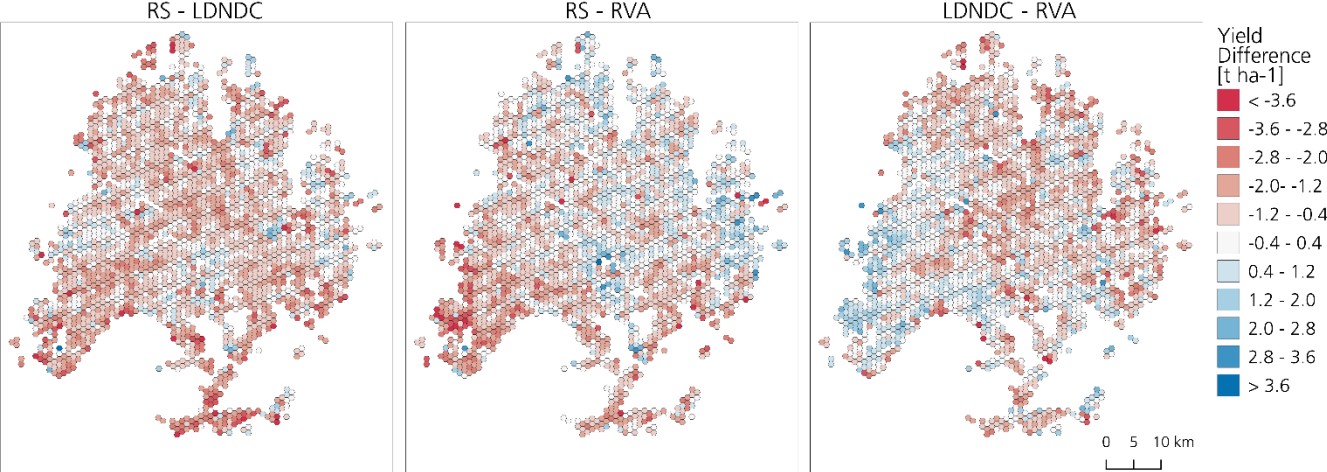

**Figure 5: Differences between spatially aggregated (hexagon diagonal length of 1km) annual yields from different models in the**
**study area in 2019. Hexagons for which the grassland area is smaller than 1 ha are not shown.**

In order to compare the estimated annual grassland yields averaged per hexagon of the three approaches, the frequency distributions were plotted together with the correlation of the annual yield estimates from the different approaches (Figure 6). The frequency distribution shows the largest range for the RS method (variance of 2.0 compared to 0.9 for LDNDC and 1.2 for RVA). The peaks showing the highest number of estimated yield ranges around 7 t ha$^{-1}$ for RS and 7.8 t ha$^{-1}$ for LDNDC

and RVA. The comparison of the hexagon yields of the RS and LDNDC approaches shows they largely overlap, in particular for the most common yield values of 6-8 t ha$^{-1}$. The Pearson's correlation coefficient is 0.67 for the hexagon yields based on RS and LDNDC. It shows significant relationships for all combinations of methods. It shows significant relationships in all combinations of methods. In general, but particularly for yields approximately below 5 t ha$^{-1}$, LDNDC shows higher values compared to the RS (and RVA) results. The comparison of annual yields averaged per hexagon from the RS and RVA

models also show a generally good agreement (Pearsons's r = 0.64), but an overall larger scattering. The hexagon yields of RVA compare well with the RS yields and also show a tendency for over- or underestimation, respectively, for smaller value ranges below approximately 6 t ha$^{-1}$ when compared to LDNDC yields (Figure 6). The relationship between RVC and LDNDC yields is the weakest with a Pearson's r of 0.47.





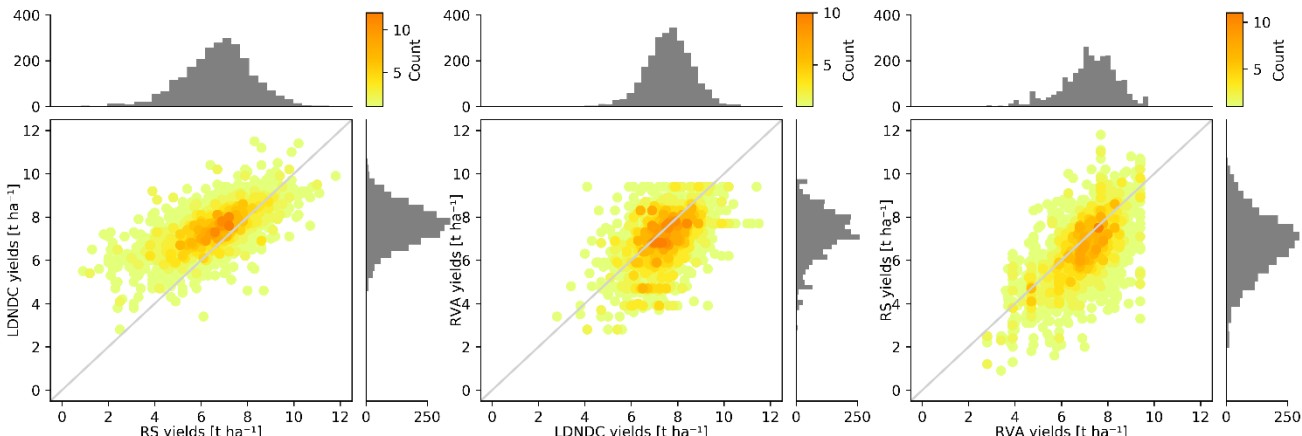

**Figure 6: Pairwise relationships of hexagon annual yields with histograms of the three modelling approaches.**

The frequency distribution and pairwise comparison of the estimated annual grassland yields per single parcel is shown in Fig. 7. The higher spatial resolution leads to increased scattering for all three approaches compared to the hexagon averages. The RS method shows the largest value range which is particularly visible from the histogram. The yields derived from RS show a variance of 4.6, 95%-percentile of 9.7 and 5%-percentile of 2.5. LDNDC and RVA yields have a variance of 2.2 and 2.7, a 95%-percentile of 9.8 and 9.4 as well as a 5%-percentile of 4.9 and 3.9. Hence, the LDNDC method covers a wider value range compared to the hexagon yield value distribution, too, but still not as wide as RS. The RVA method results in discrete values in contrast to the other two approaches and does not predict values higher than 10 t ha$^{-1}$. Due to the discrete values of RVA, there is a much higher overlap of yield values resulting in higher counts for the relationships including RVA and varying count scales in Fig. 7. As seen for the hexagon means, even though many pairs differ from the diagonal, there is a good agreement between the annual yields estimated with the RS and the LDNDC models (Pearson's r = 0.64). The largest deviation between these two model results occurs for low yields below roughly 5 t ha$^{-1}$, where LDNDC estimates higher values than RS. While the LDNDC yields shows maximum values around 12 t ha$^{-1}$, there are a few values reaching up to 15 t ha$^{-1}$ based on the RS model. The comparison of the RS and RVA results reveals a scattered pattern; however, the high number of pairs close to the diagonal in Fig. 7 indicates that there is a good agreement between the models (Pearson's r = 0.54). Similar to the RS and LDNDC relationship, RVA shows overall higher estimated yield values than RS. The LDNDC-RS relationship leads to the lowest Pearson's r of 0.44 (Figure 7). LDNDC overestimates lower yields compared to RVA and RS.



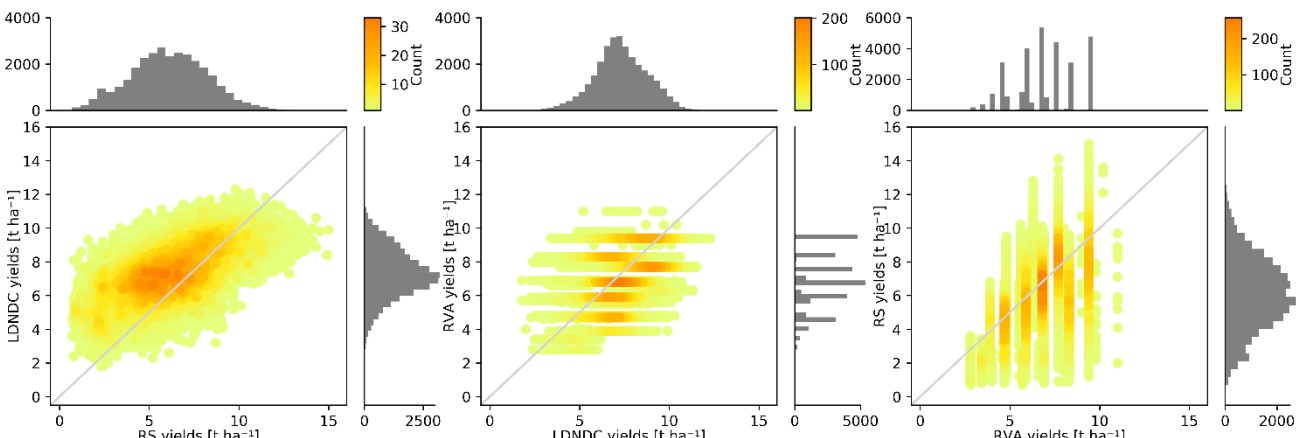

**Figure 7: Pairwise relationships of parcel-level yields with histograms of the three modelling approaches. Note, that the colour ranges differ for the subplots.**

## 4.3 Yield estimates in relation to influencing factors

### 4.3.1 Impact of mowing frequency

It can be assumed that the number of mowing and associated fertilization events per year largely influences temporal grassland vegetation growth dynamics and annual yields. We therefore investigated the estimated annual yields per mowing frequency. The RS and LDNDC models consider the same mowing dates and all approaches the same frequencies of mowing events. Boxplots of parcel-based annual yields per mowing frequency show that the estimated yield rises with the number of mowing events per year for all models (Figure 8). The mowing frequency has the strongest impact on the yield derived by the RS method, as the estimated yields show a continuous and clear increase with each additional number of annual mowing events. The Pearson's correlation coefficient, which is significant for all three approaches, is 0.81 for the number of mowing events and the RS yields, 0.74 for LDNDC and 0.66 for RVA. While the average yields for parcels mown 3-5 times correspond relatively well for all three methods, the yields for parcels mown only 1-2 times per year are lower for the RS model compared to the other two models. For a single (two-) cut field, the RS approach shows an average annual yield of 2.1 t ha$^{-1}$ (4.3 t ha$^{-1}$), whereas LDNDC predicts 4.4 t ha$^{-1}$ (6.5 t ha$^{-1}$) and RVA 4.4 t ha$^{-1}$ (5.7 t ha$^{-1}$).



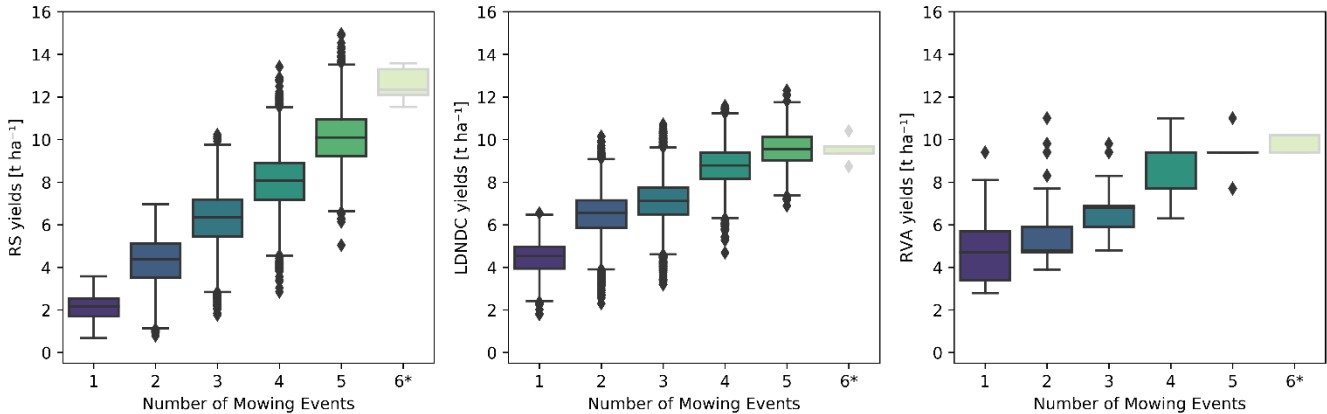

**Figure 8: Estimated annual grassland yields per mowing frequency, based on the three models RS, LDNDC and RVA. *Only six parcels were mown six times per year which might not be representative.**

### 4.3.2 Precipitation, temperature and elevation

The annual grassland yields increase with increasing mean annual temperature (MAT) mainly for RS and LDNDC but less for RVA (Figure 9), however the increase of yields is associated with a higher variability of yields at higher temperature classes. RVA yields stagnate at MAT above 12.25 °C. The Pearson's r is significant for all relationships of annual yield and MAT but shows in general a low positive correlation ((RS: r = 0.2; LDNDC: r = 0.26; RVA: r = 0.1). The yields estimated by all three methods stay relatively constant for all precipitation levels present in the study region. Pearson's r values are -0.2

(RS), -0.18 (LDNDC) and -0.05 (RVA). The relationship between yield and elevation show a negative relationship as annual yields decrease with elevation for all methods. Pearson's r for the relationships between the estimated yields and elevation are -0.19 (RS), -0.23 (LDNDC) and -0.08 (RVA). However, for the RVA yield estimates stay on average constant for an elevation of 500-900 m a.s.l. and only afterwards decline. Overall, the relationships between estimated yields and site conditions, such as temperature, precipitation and elevation, are relatively low and similar for the three methods (Figure 9).

These patterns stayed the same when the relationships were tested for individual mowing frequencies.





**Figure 9: Aggregation statistics of estimated annual yields based on the three models per temperature, precipitation and elevation classes.**




## 5 Discussion

**5.1 Performance of biomass modelling results**

Despite the numerous studies of empirical grassland biomass modelling based on satellite and field data, only few studies were carried out in areas characterized by heterogeneous and small grassland parcels which are mowed multiple times at different dates during the year. Of these previous studies the potential of various vegetation indices derived from medium resolution sensors (MODIS, MERIS) to estimate grassland biomass for single sites in Ireland and the Netherlands were

investigated (Ali et al., 2017b; Ullah et al., 2012). Based on Landsat and Sentinel-2 data, grassland biomass and height were estimated for study regions in Germany, France, Spain and Austria using various regressors, such as multi-linear regression, random forest or deep learning models, resulting in accuracies (R²) of 0.45-0.79 (Barrachina et al., 2015; Dusseux et al., 2022; Eder et al., 2023; Muro et al., 2022; Schwieder et al., 2020). The performance of empirical models seems to depend more on the number of the training data and the variety of grassland types and use intensities included than the tested

regressors and model parameters. Regarding the tested indices and bands, wetness indices (Barrachina et al., 2015) and red-edge, near-infrared and short-wave infrared bands (Dusseux et al., 2022) were found to be valuable for grassland biomass modelling. With an R² of 0.68 for the test data set, we reach comparable results with the RS approach. As far as we know, information on the time since the last mowing event was not included in previous studies so far. This parameter, however, was among the most important input features for the extreme gradient boosting model applied. Including information on

mowing dates seems to be advantageous for grassland biomass estimation in study regions characterized by intensive grassland management.

The performance of LDNDC to reproduce grassland yields of individual mowing events was found to be comparable or better than other process-based models. For the annual yields, where regional input data was employed, the performance

measures were even better than for single cuts. For further details, see Boos et al. (2024).

The comparison of the temporal patterns on estimated above-ground biomass shows that both results (RS and LDNDC) follow the mowing dynamics closely. This behaviour is expected as the mowing dates are directly included in both modelling approaches. However, the RS biomass estimates fluctuate more than the ones stemming from LDNDC and clearly

depend on the availability of cloud-free satellite observations. LDNDC biomasses show a very high peak in the first growth cycle which is often double the amount of the RS-based estimates. It is known from previous works that LandscapeDNDC overestimates yields from the first cut of the year and underestimates the ones of later cuts (Boos et al., 2024). However, it is also possible that the RS method underestimates the first cut yield as the AGB estimation might be prone to a saturation effect. In addition, AGBs at the higher end of the training data distribution are less likely predicted and strongly depend on a

well-balanced training data set.



Another important aspect is the AGB that needs to be estimated. Here, the goal was to estimate the biomass per cut, which corresponds to the yield. Farmers are generally advised to cut at 7 cm, though this can vary in practice. In the RS approach, total AGB was used because the sensor not only provides data from a certain height upwards. The LDNDC was calibrated using AGB samples taken from 7 cm. The cutting height for the reference values in the RVA approach likely varies

considerably and cannot be reconstructed. Therefore, potential uncertainties due to AGB information of different cutting heights must be assumed.

## 5.2 Spatial patterns of annual grassland yield and influencing factors

The spatial yield patterns from all three methods match in some regions and differ in others. The spatial patterns mostly resemble the mowing frequency which is a major influencing factor determining grassland yields (Bernhardt-Römermann et

al., 2011). In many cases, areas with high yield match with areas showing a higher number of mowing events. For example, in the East or the North of the study area, for many grasslands, high annual yield estimates, in particular from the LDNDC and RS models, fit well to a large number of mowing events. This relationship is also underlined by the significant Pearson correlation coefficients of 0.66-0.81 between annual yield and number of mowing events for the three models. Other regions, for example, East of lake Starnberg and in the centre of the study area show high yield estimates but rather low to

intermediate mowing frequencies (compare Figures 1 and 3). This discrepancy must be explained by other factors influencing grassland yields, which were not looked into in detail here, such as soil conditions or an optimal interplay of influencing factors. Intensified mowing, usually accompanied by more fertilization, enhances biomass production and changes species composition towards more productive vegetation with less species in systems not strongly limited by other factors (Isbell et al., 2013; Mayel et al., 2021; Savage et al., 2021). The influence of site conditions, in particular climatic

conditions, on the spatial annual yield patterns is twofold. Firstly, the conditions in the year of interest influence vegetation growth in that particular year. Secondly, climatic site conditions determine species composition and management options of grasslands in the long-term as well as soil properties like soil organic carbon. There are overall smaller yields visible in the south and southwest of the study region which matches the temperature patterns. Apart from that, the resemblance between annual yield maps and temperature, precipitation and elevation is relatively low (compare Figures 1 and 3). Pearson

correlation coefficients were significant but low for all combinations (-0.23-0.26). For correlation tests with as many data as in our case, correlations tend to be significant (Rouder et al., 2009). It can be assumed that climatic effects have a more significant influence on grassland yields on a larger spatial scale, e.g. continental scale (Emadodin et al., 2021; Goliski et al., 2018; Zhang et al., 2018). Either the climatic gradients are not large enough in our study area to explain grassland yields or climatic conditions play only a minor role as yields are mostly determined by management. A beforehand anticipated

difference in the relationship between site conditions and yields for the three models, as LDNDC includes these data directly and RS captures the effects indirectly, was not found. Furthermore, in this study only one year was examined which showed relatively normal climatic conditions. The differences between the models are most probably higher in extreme years, e.g. 2018, as extreme climatic effects can be depicted by LDNDC and RS, but not the RVA (compare Boos et al. (2024)).





### 5.3 Differences between the modelling results

When comparing the annual yield values, the RS method leads to overall lower estimates compared to LDNDC and RVA, which can be explained in several ways. On the one hand, the RS method is prone to underestimation of annual yields as the empirical model of above-ground biomass is rather underestimating. This is related to the available training data which stem from field measurements throughout the year with only a few samples from shortly before a mowing event. In this regard, the distribution of training samples might not be optimal and rather missing data of high biomass, which results in an

underestimation of the empirical model. In addition, the biomass estimation and the mowing detection are both dependent on the availability of cloud-free satellite observations. Biomass estimates from periods shortly before mowing events might be missed and, consequently, the yield related to the mowing event is potentially lower than in reality. Further, mowing events are potentially missed entirely due to cloud coverage (Reinermann et al., 2023) which additionally leads to these yields missing in the annual estimate. As the mowing information is included in all three approaches, missed mowing events due to

clouds also affects the LDNDC and RVA yields.

On the other hand, while the RS method is more likely underestimating yields, LDNDC and RVA are more prone to overestimation. This is related to the fact that some factors which negatively influence vegetation growth and, therefore, yields are not included in the LDNDC and RVA models. For LDNDC, these are neglected local factors, like north-facing

slopes and lateral run-off as well as the calibration on lysimeter data taken under favourable conditions (Haas et al., 2013). The input into the RVA model is very limited in the sense that local factors influencing the current growing conditions as well as yearly climate data are not included. Both models therefore tend to represent optimal growing conditions and hence overestimate yields. Even though the RS model is not specifically addressing these factors, they are captured by the spatially varying reflectance signal.


When examining the lower yields resulting from the RS model compared to the other model estimates, it becomes clear that this effect is most prominent for grasslands mown one or two times per year (compare Figure 8). This could be related to a grazing effect. Such extensively used grasslands are very often also grazed (Schoof et al., 2020a, c). In the LandscapeDNDC simulations, the amount of grazed material remains on the field and leads to an overestimation of the yield from the next cut,

either directly within the same year or via plant storage over the winter in the first cut of the following year. The RS method does not specifically account for grazing, and as the estimated biomass before mowing events is used, there is no such accumulation effect. Assuming that the difference between the RS and LDNDC yields of extensively used grasslands stems mostly from the grazing effect, it might be used to calculate grazed yields (e.g. Chang et al. (2015)). The RVA considers grazing on mowing pastures. However, it follows a rather simple approach estimating the grazing intensities with the farm's

stocking rate. The stocking density, i.e. the number of LSU per ha field area, would be a more meaningful measure but the IACS data does not include any information on the type of animal husbandry or grazing intensities. Therefore, yields of





mowing pastures might be over- or underestimated in the RVA resulting from lack of detail in the information on grazing intensity.

Other studies investigating grassland yield in Europe mostly focus on a continental scale and are not conducted on parcel level. However, the estimated yields are in similar value ranges compared to our results, e.g. (Chang et al., 2015; Smit et al., 2008).

## 5.4 Advantages and limitations of the approaches and implications drawn from them

All of the three approaches investigated in this study hold individual advantages and limitations to estimate annual grassland 550     yields in southern Germany. The RS approach depends on well distributed training data and cloud-free satellite observations. The value distribution of the training data determines the range of potential predictions of the empirical model and is, therefore, crucial. This also plays an important role considering the transferability (in space and time) of the approach. Without additional training data, the approach can hardly be applied in regions with different conditions. The necessity of satellite data can be particularly problematic for regions or time periods with relatively high cloud coverage. For instance, in 555     Germany, the year 2021 was characterized by an overall relatively low number of cloud-free satellite observations making satellite data-based products as the mowing detection or biomass estimation more challenging (Reinermann et al., 2023). Despite these limitations, estimating grassland yields based on RS has several advantages. Even small-scale spatial effects are depicted enabling the detection of parcels with reduced yields. Current spatio-temporal variation is mirrored, such as yield reduction through drought periods. In addition, despite the need of training data, no large input data set or 560     parametrisation is needed for the RS approach, which might even enhance the transferability.

Major advantages of LandscapeDNDC – as a bio-geochemical model – are that firstly, spatial and temporal variations are accounted for, secondly, the direct relation to input data like climate, soil, and air chemistry is possible, thirdly, carbon, nitrogen and water budgets are modelled as well, and at last, scenarios (climate or management) can be studied. This also 565     means, that high resolution input data (climate, soil, management and air chemistry) needs to be available for the modelled domains. Generally, the model performs best, if individual detailed simulations are performed, which are than aggregated to a larger spatial-temporal scale (e.g. field scale to hexagons). However, slope and orientation as well as species composition not accounted for. To transfer the model, it is ideally recalibrated based on local measurements even though for crops it has performed very well even on a global scale without this step (Jägermeyr et al., 2021). For grasslands, so far, LDNDC has 570     been used successfully in Switzerland, the UK, and Germany (Houska et al., 2017; Molina-Herrera et al., 2016; Petersen et al., 2021).

The RVA is based on field measurements and mostly static input data, apart from the mowing frequency and stocking rate. It is therefore not able to depict spatial and temporal variations, such as drought episodes (Diepolder et al., 2016). It also needs



input data, first of all the yield reference values which are rarely provided on a larger scale. Further, cattle numbers, land-use type (i.e. meadow or mowing pasture), and mowing frequencies are needed, which is – especially on scales larger than farm scale – at times difficult to obtain as it is not openly accessible or does not exist. An advantage of the RVA is that the approach is relatively straight forward and does not need large amounts of computational power. Additionally, reference values are usually used by farmers to calculate their field's fertilizer requirements. The approach is thus more useful at farm

scale where data can easily be obtained and becomes more difficult to use at larger scales due to restricted data availability.

## 6 Conclusions and outlook

Within this study, annual grassland yield was estimated for a study region in southern Germany in 2019 based on three different approaches: i) an empirical remote sensing model, ii) a bio-geochemical model (LandscapeDNDC) and iii) a rule-based reference value approach. It was shown that grassland yields can be estimated based on three completely different

approaches, as plausible and comparable results were reached for the study area. The three models contain varying input data sets, however, all use mowing information as a driver. The mowing frequency was found to be the most important influencing factor for grassland yields in the study region for all three approaches.

Yield patterns and comparisons between the approaches showed that all three methods can be legitimately used for yield

estimation (value ranges of approximately 4-9 t ha$^{-1}$) in pre-Alpine grassland ecosystems considering individual limitations. All approaches need the mowing information on parcel level as input data. When a training data set of well-balanced AGB samples and cloud-free satellite observations are available, it is advisable to use the RS approach. Depending on the training data distribution the RS approach is capable to estimate grassland yields on a regional level and also capture small-scale patterns on field level (and beyond). LandscapeDNDC is also recommended to be used on a regional or even continental

level. Detailed data on climate, soil and management is needed and strongly determines the performance. For the field scale, also the RVA can be used as the needed data can be obtained more easily for this rather than the regional level. To investigate single cut yields only RS and LDNDC can be used as RVA provides solely an annual yield estimate.

The expected biases include a likely overestimation of LDNDC of the first cut yield and an underestimation of the first cut

yield – and to a lesser degree annual yields in general – of the RS approach. To investigate yield patterns over time, only RS and LDNDC are useful as the RVA does not include actual conditions. Improved grassland yield estimations could be obtained with more AGB sample data, in particular when analysing years with climatic extremes as the AGB data might not be representative. In addition, validation data on annual grassland yields would be needed to evaluate the approaches in more detail.




The study presents synergies of grassland yield estimation approaches which is particularly important as spatial information on grassland yield is limited. Based on robust grassland yield estimations multi-annual analyses can be conducted and effects of climate change, for instance, investigated. In addition, a comprehensive understanding of grassland ecosystems is facilitated supporting authorities and science.

## Author contribution

SR, CB, AK and RK design the study. SR and AS collected the field data. SR carried out the remote sensing model, CB the process-based model and AK the reference value approach with the help from all co-authors. SR prepared the manuscript and all co-authors supported with writing or suggestions for adjustment.

## Acknowledgements

We would like to thank all state authorities which provided data and the multiple helpers during field campaigns.

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

**Financial Support**

Funding was provided by the Federal Ministry of Education and Research (BMBF) via the BonaRes project SUSALPS (031B0027A, 031B0027F).

**Data availability**

The mowing dates are provided upon request by the corresponding author.

**Competing interests**

The authors declare that they have no conflict of interest.

**Appendix**

**Table A1: Bavarian reference values for farmer's grassland yield from field measurements for meadows and mowing pastures of different use types and yield levels.**

| Use type and intensity | Farmer 's yield [t ha-1 a-1] | | |
| --- | --- | --- | --- |
| | Low | Medium | High |





| | | | |
|---|---|---|---|
| Meadow 1 cut | 2.8 | 3.4 | 4.0 |
| Meadow 2 cuts | 3.9 | 4.7 | 5.5 |
| Meadow 3 cuts | 5.6 | 6.8 | 8.0 |
| Meadow 4 cuts | 6.3 | 7.7 | 9.0 |
| Meadow 5 cuts | 7.7 | 9.4 | 11.0 |
| Meadow 6 cuts | 8.4 | 10.2 | 12.0 |
| Mowing pasture extensive, 20 % pasture | 4.8 | 5.9 | 6.9 |
| Mowing pasture medium intensive, 20 % pasture | 6.9 | 8.3 | 9.8 |
| Mowing pasture intensive, 20 % pasture | 7.7 | 9.4 | 11.0 |
| Mowing pasture extensive, 60 % pasture | 4.7 | 5.7 | 6.7 |
| Mowing pasture medium intensive, 60 % pasture | 5.7 | 6.9 | 8.1 |
| Mowing pasture intensive, 60 % pasture | 6.6 | 8.0 | 9.4 |



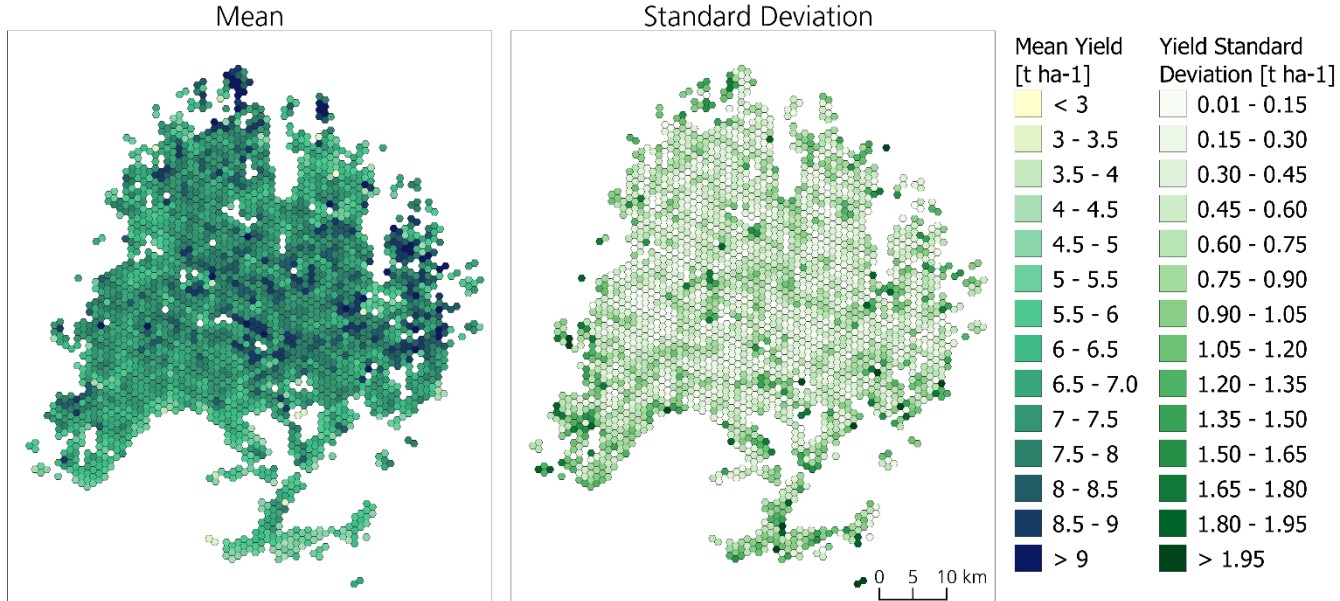

885

**Figure A2: Mean and standard deviation of spatially aggregated (hexagon diagonal length of 1km) annual yield estimates for meadows and mowing pastures in the study area in 2019 of all three approaches. Hexagons for which the grassland area is smaller than 1 ha are not shown.**