# Peer review of "Grassland yield estimations – potentials and limitations of remote sensing in comparison to process-based modelling and field measurements"

_EGUsphere, 2024_

## Author Comment (AC1)

The paper describes and compares three different approaches for grassland yield estimation. Given the limited availability of spatially explicit data on grassland yields and the pressure of global change on agricultural ecosystems, this paper addresses a relevant topic and emphasises the potential of fine-scale monitoring using high-resolution spatio-temporal earth observation and environmental data.

The main novelty of this paper seems to me to be the presentation of a novel remote sensing-based approach that combines modelled AGB time series and mowing events, all derived from free and globally available satellite data. This approach is compared with an established process-based model (LDNDC) and a simple rule-based approach. The paper is well written and clear.

I recommend it for publication but encourage the authors to make some improvements to clarify and increase its contribution to a larger audience. My main criticism is expressed in the first comment, which suggests strengthening the focus and improving the results section.

*Thank you for taking the time to review our manuscript as well as for this positive assessment and critique. It is correct that the remote sensing-approach is novel and we would like to follow your suggestion of putting more emphasize on the remote sensing results along with other important remarks made (more below). We appreciate the clear and constructive comments of the reviewer and think that the suggested changes will improve the manuscript. We answered all comments below in italic while the reviewer remarks are upright.*

**General comments**

1) I think the relevance of this paper could be increased with a stronger focus on the novel RS-based approach and its ability to capture small-scale spatial and temporal variations, with management information being a prerequisite, also for LDNDC. The importance of spatial/temporal variation is highlighted in the abstract and introduction, and its potential to be captured by the RS-based approach is also highlighted in the conclusion under L593/594, but in my opinion, this is not adequately represented in the results. I recommend a more detailed analysis of this.

*Thanks for this assessment. We agree with this and would like to follow your suggestions made below.*

A stronger focus on the RS-based approach would make it clearer what the novelty of this paper is, especially as the other two approaches are being currently under review in a separate paper, if I understood correctly. In this regard, it might be worth shortening the description of LDNDC in the methods and the detailed and rather descriptive comparison with the two other approaches (results and discussion section, e.g. L376-L407), which I think are interchangeable to some extent. This could be complemented with some in-depth analysis of the novel RS-based approach. For instance, I think the currently made statements such as "All three approaches reach plausible results of annual yields of around 4-9 t/ha and show overlapping as well as diverging spatial patterns" are not very meaningful. I'm also not sure how relevant this is for people who don't work with LDNDC, or if it allows general conclusions to be drawn for process-based models?

*Thank you for these suggestions. We agree with the shift of the focus towards the RS approach. Therefore, we will expand the RS-based results and shorten other parts of the manuscript, in particular the method description of LDNDC and some rather descriptive comparisons. The rule-based approach is just a small part of a larger agent-based modelling of ecosystem services in the region. In the meantime, the paper to this ecosystem service modeling approach was published (https://doi.org/10.1016/j.eja.2025.127539). However, the focus of this paper is not the yield modelling but the nitrogen fertilizer input, while the modeled yield is rather an intermediate product. The LDNDC yield modelling paper is probably resubmitted soon and not published yet. However, here the focus lies on the effect of drought conditions and a comparison of multiple years, also looking at the results with a different scale. Therefore, we think it is still necessary to present the results of the other two approaches as well. However, we will shorten the mentioned parts of the manuscript to reduce the repetitiveness. We assume that the results of LDNDC can be transferred to other process-based models (e.g. Daycent, APSIM). We will add this to the discussion.*

Can the authors show (map, stats) that small-scale variations are better captured with the RS-based approach (e.g. looking at adjacent in-situ measurements / within and across parcel variations)? Same for the temporal variation, e.g. which approach better captures AGB of the first cut (see statement L26/L470; it's not clear to me, was it compared with in-situ measurements?)? I think these points are rather well covered in the discussion but could be improved in the results section. Also consider changing the title to give more weight to the RS-based approach and clearly mention the focus in the introduction.

*Thank you for this important remark. We highly favor the suggestions made to highlight small-scale variations captured by the RS-based approach. It is indeed an advantage and not presented well enough in the current version of the manuscript. All three approaches result in yields per parcel. However, the RS-based approach considers within-parcel variations as biomasses per pixel are calculated. Therefore, we would like to add a map showing the estimated biomass for several time steps to highlight this. We will add the LDNDC biomass in a numeric way as the data is only available per parcel and we are not allowed to show parcel boundaries due to data restrictions. In addition, we will add the adjacent in-situ measurements in Figure 3 to enable a temporal comparison of results as well. We are open to change the title of the manuscript and suggest "Grassland yield estimations – potentials and limitations of remote sensing in comparison to process-based modelling and field measurements".*

If the authors choose to keep their focus on comparing the three approaches, more in-depth analysis is needed, e.g. to clearly understand why one model over/underestimates yield and under what conditions. This should go beyond "simply" describing differences at hexagon-level and frequency distributions in the result section.

*We decided to follow the suggestion of moving the focus more towards the RS-based results and agree that otherwise a more in-depth analysis of the comparison would be needed.*

In the first chapter of the results, please provide more details about the RS-based model, e.g. prediction vs. observed plot.

*Thank you for this suggestion. As suggested we will extend the first chapter of the results with more detailed results of the RS-based approach. A prediction vs. observed plot is a well-suited option to do so. We will include it in the revised version of the manuscript as well as*

*further more detailed results, such as a map of temporal biomass estimates showing within-parcel variability as described above.*

2) More information regarding the in-situ measurements is needed (section biomass field data). In my opinion it is not sufficient to simply provide a reference, at least an overview of the sampling design is needed, addressing the nested design, the spatial and temporal distribution, and the number of samples per mowing frequency.

*Thank you for this remark as well as the specific and helpful suggestions. We agree that the description of the in-situ measurements was relatively short. In an updated version of the manuscript we will extend the description, including the mentioned information.*

3) The results focus on showing the differences between the three approaches (map, histogram). However, these are difficult to interpret in terms of which approach performs better. A more detailed comparison/analysis with in-situ measurements would give more substance to this paper. However, I'm not sure if the in-situ data allows such further analysis? If so, this could help to address questions raised under comment #1, is linked to specific comments bellow and the potentially missing in-situ symbols in figure 3.

*Thank you for this comment. It is correct that the in-situ data is limited and might not allow for a reliable comparative analysis. However, the in-situ measurement symbols in Figure 3 were indeed missing, we apologize for this mistake. We hope that adding the symbols there will improve the interpretability of the plausibility of the approaches and their differences.*

4) The comparison of the RS-based model with LDNDC is a bit difficult to interpret, as both approaches rely on Sentinel-2-based mowing events. Please clarify what this means for the interpretation, e.g. what would be the performance of LDNDC without spatially explicit mowing information (or is there another data source for this)?

*Thank you for this remark. It is correct that all three of the approaches rely on the same mowing dates dataset. This might make it harder to differentiate the results but also provides the same baseline for the three approaches. All three approaches rely on the mowing dates and don't work without. For the LDNDC model, different solutions without a RS-based mowing dates dataset exist, for example using simulated mowing dates depending on standing biomass as described in Krischan et al. (2021) (*https://doi.org/10.1016/j.eja.2021.126306*). However, the missing mowing dates were usually seen as a disadvantage of models such as LDNDC in the past. We will add this aspect in the discussion to make the reader more aware of the circumstances.*

As both models are highly dependent on the RS-based mowing dates, the finding that the spatial patterns or box plots (Figure 8) between the two approaches are similar and strongly related to the number of mowing events is somewhat redundant/obvious (e.g. L418, L467). I don't know how this could be resolved, perhaps by running separate models with/without mowing events?

*Thank you for this comment. We agree that it seems straightforward that the RS and LDNDC approaches are strongly related to the mowing events. However, with Figure 8 we would like to show how the influence of the mowing frequency might differ between all three yield estimation approaches. For example, we found that the RS-based approach shows lower yields for low intensity grasslands and higher values for high intensity ones. We will review the text to make sure that the results are formulated in a clear manner. The difficulty with*

*running separate models, either with or without mowing data is, that the models need the mowing dates as input as there are no yields without harvesting. It is therefore not possible to use these approaches without mowing information.*

5) Please review figure and table captions to be more self-explaining and consistent (e.g. Figure 4 vs. Figure 5 vs. Figure 6; rather mention the different modelling approach like in Fig. 4) and use consistent terms in the whole document (e.g. LandscapeDNDC vs. LDNDC).

*Thank you for these remarks. We will adjust the figure and table captions to make them more informative and consistent and will revise the manuscript to use consistent terms in the entire document.*

**Specific comments**

L21: Change "present" with "compare", or clarify which approach is new (RS-based) and which already exist / will be published separately?

*Thanks, we will clarify that.*

L140 /Figure 1: Change to "CLCplus Backbone 2021", consider citation, remove not-shown legend items. Make clear that the study area is the Ammer catchment and add catchment border to sub-figures.

*We will follow the suggestions and make the according changes.*

L155: Was the validation done separately for this study or is the F1-score derived from Reinermann et al.? More details or citation is needed. Please check that the uncertainty of mowing detection is adequately addressed in the discussion section, as this is the most important parameter.

*Thank you for highlighting this. It is the same validation approach and based on the same data but in fact a separate validation as only the subset of data from the Ammer region was used. We will add this information in the methods and check that the uncertainty of the mowing detection is adequately discussed.*

L166: Add plot locations to the map in figure 1 (maybe top-left)?

*That is a good idea. We will do so.*

L170/171: More information regarding the sampling design is needed, I cannot quite follow how the n=111 samples are reached

*Thanks for the remark. We will revise the description of the in-situ data.*

L178 (Figure 2): Add validation procedure?

*We will think about that; however, we are not sure if it fits well.*

L180: Title not clear, consider "Remote sensing-based approach"

*Thanks for pointing us to that. We will revise the titles of the sub-headlines.*

L195: Please check EVI formula for correctness (position of factor 2.5 and apostrophe after 1)

*Thank you for highlighting this. It is correct, that the factor 2.5 should be positioned in front of the fraction line. The apostrophe is a comma after the formula. We will try make sure during the editing that this doesn't look confusing.*

L205: Please cite the HR-VPP (https://doi.org/10.2909/c1c46cb2-b02b-4013-aae5-a54a8c018b1e)

*Thank you for the source. We will cite it.*

L220: It would be helpful to get some information about the distribution of these data pairs, e.g. a figure with y=field plots, x=dates S2&field, colour=S2/field/pairs(train/test), table (supplementary) or at least providing some more information in the text (how many field plots etc.)?

*Thank you for the comment. We will add information to the data pairs, either as a table or as additional text.*

L244: Introduce r2 instead of L247; use PRMSE like in L248

*We will revise the sentences and introduce r2 when it is mentioned first and correct the abbreviations which seem to be confused here.*

L282: It sounds like the RVA is based on an official table, but then it gets more complicated with the tables and the link to Kaim et al. (under review). It is a bit confusing, please check/revise.

*We will revise the RVA description in the revised manuscript.*

L286: Do you mean "Table A1"?

*Yes, thanks for pointing to this. We will change "Appendix A1" to "Table A1".*

L311: Improve caption. Maybe move table to appendix, to make clear that it belongs to Table A1.

*We will advise the caption and think about moving it to the Appendix as you are right that it belongs to Table A1 and might be difficult to interpret without Table A1 anyways.*

L335 (Figure 3): I cannot find the in-situ measurements! Provide annual values (which are a bit hidden) as stacked-bar adding AGB from the individual events?

*Thank you for this important remark. The in-situ measurement points are missing from the Figure. We apologize for this mistake. We will add them in the revised version and will try out to add stacked bars of the event AGBs and see if the figure remains clear.*

L342: 0.97 seems to indicate overfitting and might need to be discussed later?

*It could indeed indicate overfitting. We will add some discussion to this.*

L389 (Figure 6): I suggest adding r2 and variance to the plots. Same for Figure 7.

*Thanks for the suggestion, we will add this to the figures.*

L433: Remove one bracket

*Thanks, we will do so.*

L518: You could mention the potential to tackle cloud coverage with SAR

*Yes, that is right. We will mention this here in the revised manuscript.*

L550: In this study LDNDC also depends on cloud-free observations for the mowing dates

*That is correct. We will mention this as well.*

---

## Author Comment (AC2)

Grasslands make up most agricultural land and provide fodder for livestock, yet data on grassland yields is limited due to direct use on farms. Accurate yield information is crucial for informing policy and understanding ecosystem services and inter-annual variations. In this study, we estimate annual grassland yields in the Ammer catchment area of southern Germany for 2019 using three approaches: (i) a model combining field samples, satellite data, and mowing information (RS), (ii) the biogeochemical process-based model LandscapeDNDC (LDNDC), and (iii) a rule-based approach based on field measurements and spatial productivity data (RVA). All approaches yield similar results, estimating yields of 4-9 t/ha, with some spatial variations. LDNDC generally produces higher yields, especially for the first cut and grasslands mown one or two times per year. Mowing frequency was the most significant factor influencing yields, with no major differences in the impact of abiotic factors (e.g., climate or elevation) across approaches. This comparison offers new insights into the strengths and limitations of each method and highlights the importance of grassland productivity maps for long-term studies on climate and management impacts.

This study is a pioneering effort to compare entirely different approaches for estimating grassland biomass and provides valuable insights for future land use estimations. I recommend it for publication, with minor to moderate revisions focusing on enhancing clarity and broadening its relevance for a wider audience.

*We would like to thank you for taking the time and reviewing our manuscript. We appreciate the positive feedback and comments. We are confident that the revisions made in the manuscript based on the suggestions of the reviewer will enhance its quality. We answer to each comment below in italics while the reviewer's comments are upright.*

General comments:
**1 The key findings highlight the spatial and temporal differences between the three approaches. However, these differences are difficult to interpret in terms of the underlying reasons why each model or approach yields distinct results, and it is unclear which approach performs better overall. The results section presents only the spatial and temporal disparities without delving into potential explanations. Factors such as climate forcing uncertainties in the LDNDC model input, the quality of data, methods used for remote sensing photo analysis, and limitations of survey data could all influence the outcomes.**

To enhance the clarity and depth of the analysis, I recommend the following:

1. A more comprehensive comparison, either in a table or in the discussion section, outlining the pros and cons of each method.
2. An in-depth analysis that investigates the potential root causes behind the observed differences in results, including an exploration of the most significant factors affecting each approach (beyond mowing).
3. A discussion on the future implementation of these approaches, specifying under which conditions each model is more suitable than the others.

*Thank you for these suggestions. Regarding your first point, we agree that a table would improve the clarity and show pros and cons in a concise manner. Such a table would be rather qualitative as the limited number of in-situ data doesn't allow for a quantitative in-depth assessment, unfortunately. We would still like to follow this idea and add a table with pros and cons to the revised version of the manuscript, most probably in the conclusions.*

*Regarding your second point, we think that such an in-depth analysis of influencing factors would be very interesting. We analyzed the influence of the most relevant factors for grassland yield, namely mowing frequency, temperature, precipitation and elevation (compare chapter 4.3) by assessing the correlation between these factors and the estimated yields and the plots. We hope that the results are formulated clearly but will revise the manuscript to make sure that our findings are well understandable. We suggest that we could add additional plots as Figure 9 and statistics between influencing factors and yields per mowing frequency (for all or only the most dominant mowing frequencies). In that regard the influence of temperature, precipitation and elevation is investigated apart from the mowing frequency. However, we checked the results and they are very similar to the Figure 9.*

*Regarding your third point, we agree that the discussion so far lacks specific suggestions for future implications. We will add this to the discussion as far as it is possible.*

**2 Each method, due to its underlying model mechanism and input data, introduces uncertainties. However, the study presents only the estimated annual AGB from the three approaches without providing an uncertainty range for each estimation. To draw a more reliable conclusion about the similarity of the three results, it is essential to include upper and lower bounds for each estimate. I recommend incorporating the potential uncertainties associated with the input data for each method and providing confidence intervals (upper and lower bounds) for the annual AGB estimations.**

*Thank you for this remark. We agree that it is important to include uncertainty information to modelled results. For the RS and LDNDC approaches the validations result in r2 and RMSE (root mean square error) estimates, which are described in L340 and L345. To analyze the sensitivity of the models towards the input data would indeed give a more sophisticated insight into model uncertainties. However, there are no uncertainties related to all input datasets, hindering us from applying an error propagation analysis. In addition, this seems to be out of scope for this manuscript. We hope that providing the estimated errors derived from the validation based on in-situ measurements is sufficient. We will add error bars to the annual yield estimates of Figure 3 to highlight these.*

**3 The results suggest that all approaches are highly sensitive to mowing dates, with the estimated annual grassland yields showing a strong correlation to the number of mowing events. To assess the stability and effectiveness of each model, I recommend conducting a sensitivity test by removing the number of mowing events from all three models. This would allow for an evaluation of how each model performs with the remaining input variables**

*Thank you for highlighting this aspect and the suggestion to test this. We agree that the idea to test how the models perform without mowing data sounds interesting as this seems to be the most important input variable. However, all approaches rely on the mowing dates as input. There are no yield estimates without harvesting information. Therefore, we can't test this using these approaches and our model setups, unfortunately. However, as mentioned above we could add additional plots to the Appendix showing the relationships of yields to temperature, precipitation and elevation per mowing frequency.*

**4 The annual grassland biomass estimation is based on a particular region in this study. It would be better to give some insights in the discussion section to show the possibility and potential use case to apply those three approaches to a bigger region, what the limitations (such as data availability) would be, and what method(s) will be most likely to serve better.**

*Thanks for this important comment. In the current version, we missed discussing the transferability of the approaches to other regions. We will add this to the discussion in the revised manuscript.*

Specific comments:

**1 Figure 3. Related to my comment #2, try to add an uncertainty range for the annual AGB**

*Thank you for this suggestion. We will add the estimated errors to the annual yields of Figure 3 to show uncertainty ranges.*

**2 Figure 5. Try to reverse the color bar by using red as positive and blue as negative**
*Thanks for this comment. We will follow your suggestion and change the colors to make the plot more intuitive.*

**3 Line 430 - 440 & Figure 9. Is the correlation between annual yields and temperature, precipitation, and elevation obtained without dropping dominant mowing events? If so consider eliminating mowing events from the model and test the sensitivity**

*Thank you for your remark. The mowing events were not dropped here. But we tested the relationships per mowing frequency (not shown in current manuscript version yet). The results stayed relatively the same. Nevertheless, as suggested above, we could add the Figures and statistics per mowing frequency to show that the relationships stay constant among varying mowing frequencies.*

---

## Author Response (AR1)

Rev #1

The paper describes and compares three different approaches for grassland yield estimation. Given the limited availability of spatially explicit data on grassland yields and the pressure of global change on agricultural ecosystems, this paper addresses a relevant topic and emphasises the potential of fine-scale monitoring using high-resolution spatio-temporal earth observation and environmental data.

The main novelty of this paper seems to me to be the presentation of a novel remote sensing-based approach that combines modelled AGB time series and mowing events, all derived from free and globally available satellite data. This approach is compared with an established process-based model (LDNDC) and a simple rule-based approach. The paper is well written and clear.

I recommend it for publication but encourage the authors to make some improvements to clarify and increase its contribution to a larger audience. My main criticism is expressed in the first comment, which suggests strengthening the focus and improving the results section.

*Thank you for taking the time to review our manuscript as well as for this positive assessment and critique. It is correct that the remote sensing-approach is novel and we would like to follow your suggestion of putting more emphasize on the remote sensing results along with other important remarks made (more below). We appreciate the clear and constructive comments of the reviewer and think that the suggested changes improved the manuscript. We answered all comments below in italic while the reviewer remarks are upright.*

**General comments**

1) I think the relevance of this paper could be increased with a stronger focus on the novel RS-based approach and its ability to capture small-scale spatial and temporal variations, with management information being a prerequisite, also for LDNDC. The importance of spatial/temporal variation is highlighted in the abstract and introduction, and its potential to be captured by the RS-based approach is also highlighted in the conclusion under L593/594, but in my opinion, this is not adequately represented in the results. I recommend a more detailed analysis of this.

*Thanks for this assessment. We agree with this and would like to follow your suggestions made below.*

A stronger focus on the RS-based approach would make it clearer what the novelty of this paper is, especially as the other two approaches are being currently under review in a separate paper, if I understood correctly. In this regard, it might be worth shortening the description of LDNDC in the methods and the detailed and rather descriptive comparison with the two other approaches (results and discussion section, e.g. L376-L407), which I think are interchangeable to some extent. This could be complemented with some in-depth analysis of the novel RS-based approach. For instance, I think the currently made statements such as "All three approaches reach plausible results of annual yields of around 4-9 t/ha and show overlapping as well as diverging spatial patterns" are not very meaningful. I'm also not sure how relevant this is for people who don't work with LDNDC, or if it allows general conclusions to be drawn for process-based models?

*Thank you for these suggestions. We agree with the shift of the focus towards the RS approach. We adapted the introduction to highlight the focus on the remote sensing approach (L100ff, L120ff). We expanded the RS-based results (see below) and shortened other parts of the manuscript, in particular the method description of LDNDC (Section 3.2.2) and some rather descriptive comparisons (L414-430). The rule-based approach is just a small part of a larger agent-based modelling of ecosystem services in the region. In the meantime, the paper to this ecosystem service modeling approach was published (*[https://doi.org/10.1016/j.eja.2025.127539](https://doi.org/10.1016/j.eja.2025.127539)*). However, the focus of this paper is not the yield modelling but the nitrogen fertilizer input, while the modeled yield is rather an intermediate product. The LDNDC yield modelling paper will probably be resubmitted soon but is not published yet. However, here the focus lies on the effect of drought conditions and a comparison of multiple years, also looking at the results with a different scale. Therefore, we think it is still necessary to present the results of the other two approaches as well to a certain degree. However, we shortened the mentioned parts of the manuscript to reduce the repetitiveness, e.g. LDNDC method description (section 3.2.2.). We assume that the results of LDNDC can be transferred to other process-based models (e.g. Daycent, APSIM). We will add this to the discussion (L490-491).*

Can the authors show (map, stats) that small-scale variations are better captured with the RS-based approach (e.g. looking at adjacent in-situ measurements / within and across parcel variations)? Same for the temporal variation, e.g. which approach better captures AGB of the first cut (see statement L26/L470; it's not clear to me, was it compared with in-situ measurements?)? I think these points are rather well covered in the discussion but could be improved in the results section. Also consider changing the title to give more weight to the RS-based approach and clearly mention the focus in the introduction.

*Thank you for this important remark. We highly favor the suggestions made to highlight small-scale variations captured by the RS-based approach. It is indeed an advantage and was not presented well enough in the previous version of the manuscript. All three approaches result in yields per parcel. However, the RS-based approach considers within-parcel variations as biomasses per pixel are calculated. Therefore, we included a map showing the estimated biomass for several time steps to highlight this (Figure 3). Unfortunately, information per parcel cannot be shown as we are not allowed to show parcel boundaries due to data restrictions. In addition, we added the adjacent in-situ measurements in Figure 5 (former Figure 3) to enable a temporal comparison of results as well. We are open to change the title of the manuscript and suggest "Grassland yield estimations – potentials and limitations of remote sensing in comparison to process-based modelling and field measurements".*

If the authors choose to keep their focus on comparing the three approaches, more in-depth analysis is needed, e.g. to clearly understand why one model over/underestimates yield and under what conditions. This should go beyond "simply" describing differences at hexagon-level and frequency distributions in the result section.

*We decided to follow the suggestion of moving the focus more towards the RS-based results.*

In the first chapter of the results, please provide more details about the RS-based model, e.g. prediction vs. observed plot.

*Thank you for this suggestion. As suggested we extended the first chapter of the results with more detailed results of the RS-based approach (Figure 3, LL340ff) and created a separate chapter for it. A prediction vs. observed plot was included as well (Figure 4).*

2) More information regarding the in-situ measurements is needed (section biomass field data). In my opinion it is not sufficient to simply provide a reference, at least an overview of the sampling design is needed, addressing the nested design, the spatial and temporal distribution, and the number of samples per mowing frequency.

*Thank you for this remark as well as the specific and helpful suggestions. We agree that the description of the in-situ measurements was relatively short. In the revised version of the manuscript we added more information, such as to the nested design (L 173-174), to the spatial and temporal distribution of the sampling data (Figure 1 and L 178-181) and to the number of samples per mowing frequency (L 178-181).*

3) The results focus on showing the differences between the three approaches (map, histogram). However, these are difficult to interpret in terms of which approach performs better. A more detailed comparison/analysis with in-situ measurements would give more substance to this paper. However, I'm not sure if the in-situ data allows such further analysis? If so, this could help to address questions raised under comment #1, is linked to specific comments bellow and the potentially missing in-situ symbols in figure 3.

*Thank you for this comment. It is correct that the in-situ data is limited and might not allow for a reliable comparative analysis. However, the in-situ measurement symbols in Figure 5 (former Figure 3) were indeed missing, we apologize for this mistake. We hope that adding the symbols there improves the interpretability of the plausibility of the approaches and their differences.*

4) The comparison of the RS-based model with LDNDC is a bit difficult to interpret, as both approaches rely on Sentinel-2-based mowing events. Please clarify what this means for the interpretation, e.g. what would be the performance of LDNDC without spatially explicit mowing information (or is there another data source for this)?

*Thank you for this remark. It is correct that all three of the approaches rely on the same mowing dates dataset. This might make it harder to differentiate the results but also provides the same baseline for the three approaches. All three approaches rely on the mowing dates and don't work without. For the LDNDC model, different solutions without a RS-based mowing dates dataset exist, for example using simulated mowing dates depending on standing biomass as described in Krischan et al. (2021) (*https://doi.org/10.1016/j.eja.2021.126306*). However, the missing mowing dates were usually seen as a disadvantage of models such as LDNDC in the past. We included this aspect in the discussion to make the reader more aware of the circumstances (LL 571-576).*

As both models are highly dependent on the RS-based mowing dates, the finding that the spatial patterns or box plots (Figure 8) between the two approaches are similar and strongly related to the number of mowing events is somewhat redundant/obvious (e.g. L418, L467). I don't know how this could be resolved, perhaps by running separate models with/without mowing events?

*Thank you for this comment. We agree that it seems straightforward that the RS and LDNDC approaches are strongly related to the mowing events. However, with Figure 8 we would like*

*to show how the influence of the mowing frequency might differ between all three yield estimation approaches. For example, we found that the RS-based approach shows lower yields for low intensity grasslands and higher values for high intensity ones. We reviewed the text to make sure that the results are formulated in a clear manner (LL 441). The difficulty with running separate models, either with or without mowing data is, that the models need the mowing dates as input as there are no yields without harvesting. It is therefore not possible to use these approaches without mowing information.*

5) Please review figure and table captions to be more self-explaining and consistent (e.g. Figure 4 vs. Figure 5 vs. Figure 6; rather mention the different modelling approach like in Fig. 4) and use consistent terms in the whole document (e.g. LandscapeDNDC vs. LDNDC).

*Thank you for these remarks. We will adjust the figure and table captions to make them more informative and consistent and will revise the manuscript to use consistent terms in the entire document. It says LandscapeDNDC in the figure captions as it is the full name of the model and the other two approaches are also stated with their full names in contrast to the abbreviations used in the text. We hope that this is fine.*

**Specific comments**

L21: Change "present" with "compare", or clarify which approach is new (RS-based) and which already exist / will be published separately?

*Thanks, we changed the wording as suggested (L23) and added the information which approach is novel also in the next sentence (L24).*

L140 /Figure 1: Change to "CLCplus Backbone 2021", consider citation, remove not-shown legend items. Make clear that the study area is the Ammer catchment and add catchment border to sub-figures.

*We followed the suggestions and revised Figure 1. We changed the legend header, removed not-shown items, added the plot locations (see comment below) and refined the study area to make clear where it is.*

L155: Was the validation done separately for this study or is the F1-score derived from Reinermann et al.? More details or citation is needed. Please check that the uncertainty of mowing detection is adequately addressed in the discussion section, as this is the most important parameter.

*Thank you for highlighting this. It is the same validation approach and based on the same data but in fact a separate validation as only the subset of data from the Ammer region was used (L 163-164). We added this information in the methods and mentioned the uncertainty of the mowing detection in the discussion (L 574ff).*

L166: Add plot locations to the map in figure 1 (maybe top-left)?

*That is a good idea. We included the plot locations in Figure 1.*

L170/171: More information regarding the sampling design is needed, I cannot quite follow how the n=111 samples are reached

*Thanks for the remark. We revised the description of the in-situ data (LL 179-182).*

L178 (Figure 2): Add validation procedure?

*We decided against this as we felt that it would not fit.*

L180: Title not clear, consider "Remote sensing-based approach"

*Thanks for pointing us to that. We revised the titles of the sub-headlines.*

L195: Please check EVI formula for correctness (position of factor 2.5 and apostrophe after 1)

*Thank you for highlighting this. It is correct, that the factor 2.5 should be positioned in front of the fraction line. The apostrophe is a comma after the formula. We will try make sure during the editing that this doesn't look confusing.*

L205: Please cite the HR-VPP (https://doi.org/10.2909/c1c46cb2-b02b-4013-aae5-a54a8c018b1e)

*Thank you for the source. We added it.*

L220: It would be helpful to get some information about the distribution of these data pairs, e.g. a figure with y=field plots, x=dates S2&field, colour=S2/field/pairs(train/test), table (supplementary) or at least providing some more information in the text (how many field plots etc.)?

*Thank you for the comment. We added more information in the description of the biomass samples and added a prediction vs. observation plot in the Appendix to provide more information on the data.*

L244: Introduce r2 instead of L247; use PRMSE like in L248

*We revised the sentences and introduced r2 when it is mentioned first and corrected the abbreviations which seem to be confused here.*

L282: It sounds like the RVA is based on an official table, but then it gets more complicated with the tables and the link to Kaim et al. (under review). It is a bit confusing, please check/revise.

*We revised the RVA description paragraph and hope it is clearer now (section 3.2.3).*

L286: Do you mean "Table A1"?

*Yes, thanks for pointing to this. We changed "Appendix A1" to "Table A1".*

L311: Improve caption. Maybe move table to appendix, to make clear that it belongs to Table A1.

*We revised the caption and moved the table to the Appendix as the interpretability of the RVA approach is enhanced with both tables together.*

L335 (Figure 3): I cannot find the in-situ measurements! Provide annual values (which are a bit hidden) as stacked-bar adding AGB from the individual events?

*Thank you for this important remark. The in-situ measurement points were missing from the Figure. We apologize for this mistake. We included them in the revised version and added error bars to the annual yields (Figure 5, former Figure 3).*

L342: 0.97 seems to indicate overfitting and might need to be discussed later?

*It could indeed indicate overfitting. We hinted to this in the discussion (L 482).*

L389 (Figure 6): I suggest adding r2 and variance to the plots. Same for Figure 7.

*Thanks for the suggestion, we added the information in the text.*

L433: Remove one bracket

*Thanks, we removed the bracket.*

L518: You could mention the potential to tackle cloud coverage with SAR

*We didn't include this here as we fear this is not in the scope anymore but mentioned the uncertainty of the mowing detection.*

L550: In this study LDNDC also depends on cloud-free observations for the mowing dates

*That is correct. This information is included in the discussion (L545-546).*

Rev #2

Grasslands make up most agricultural land and provide fodder for livestock, yet data on grassland yields is limited due to direct use on farms. Accurate yield information is crucial for informing policy and understanding ecosystem services and inter-annual variations. In this study, we estimate annual grassland yields in the Ammer catchment area of southern Germany for 2019 using three approaches: (i) a model combining field samples, satellite data, and mowing information (RS), (ii) the biogeochemical process-based model LandscapeDNDC (LDNDC), and (iii) a rule-based approach based on field measurements and spatial productivity data (RVA). All approaches yield similar results, estimating yields of 4-9 t/ha, with some spatial variations. LDNDC generally produces higher yields, especially for the first cut and grasslands mown one or two times per year. Mowing frequency was the most significant factor influencing yields, with no major differences in the impact of abiotic factors (e.g., climate or elevation) across approaches. This comparison offers new insights into the strengths and limitations of each method and highlights the importance of grassland productivity maps for long-term studies on climate and management impacts.

This study is a pioneering effort to compare entirely different approaches for estimating grassland biomass and provides valuable insights for future land use estimations. I recommend it for publication, with minor to moderate revisions focusing on enhancing clarity and broadening its relevance for a wider audience.

*We would like to thank you for taking the time and reviewing our manuscript. We appreciate the positive feedback and comments. We are confident that the revisions made in the manuscript based on the suggestions of the reviewer will enhance its quality. We answer to each comment below in italics while the reviewer's comments are upright.*

General comments:
**1 The key findings highlight the spatial and temporal differences between the three approaches. However, these differences are difficult to interpret in terms of the underlying reasons why each model or approach yields distinct results, and it is unclear which approach performs better overall. The results section presents only the spatial and temporal disparities without delving into potential explanations. Factors such as climate forcing uncertainties in the LDNDC model input, the quality of data, methods used for remote sensing photo analysis, and limitations of survey data could all influence the outcomes.**

To enhance the clarity and depth of the analysis, I recommend the following:

1. A more comprehensive comparison, either in a table or in the discussion section, outlining the pros and cons of each method.
2. An in-depth analysis that investigates the potential root causes behind the observed differences in results, including an exploration of the most significant factors affecting each approach (beyond mowing).
3. A discussion on the future implementation of these approaches, specifying under which conditions each model is more suitable than the others.

*Thank you for these suggestions. Regarding your first point, we agree that a table would improve the clarity and show pros and cons in a concise manner. Such a table would be rather qualitative as the limited number of in-situ data doesn't allow for a quantitative in-depth assessment, unfortunately. We still followed this idea and added a table with pros and cons to the revised version of the manuscript at the end of the discussion (Table 1, L 622).*

*Regarding your second point, we think that such an in-depth analysis of influencing factors would be very interesting. We analyzed the influence of the most relevant factors for grassland yield, namely mowing frequency, temperature, precipitation and elevation (compare chapter 4.3) by assessing the correlation between these factors and the estimated yields and the plots. We hope that the results are formulated clearly but revised the manuscript to make sure that our findings are well understandable. We added additional plots as Figure 10 (former Figure 9) to show the relationship of climate to yields per mowing frequency. In that regard the influence of temperature, precipitation and elevation is investigated apart from the mowing frequency. However, we checked the results and they are very similar to the Figure 10.*

*Regarding your third point, we agree that the discussion so far lacks specific suggestions for future implications. We added this to the discussion as far as possible (compare LL 614-621 and Table 1).*

**2 Each method, due to its underlying model mechanism and input data, introduces uncertainties. However, the study presents only the estimated annual AGB from the three approaches without providing an uncertainty range for each estimation. To draw a more reliable conclusion about the similarity of the three results, it is essential to include upper and lower bounds for each estimate. I recommend incorporating the potential uncertainties**

associated with the input data for each method and providing confidence intervals (upper and lower bounds) for the annual AGB estimations.

*Thank you for this remark. We agree that it is important to include uncertainty information to modelled results. For the RS and LDNDC approaches the validations result in r2 and RMSE (root mean square error) estimates, which are described in L363 and L368. To analyze the sensitivity of the models towards the input data would indeed give a more sophisticated insight into model uncertainties. However, there are no uncertainties related to all input datasets, hindering us from applying an error propagation analysis. In addition, this seems to be out of scope for this manuscript. We hope that providing the estimated errors derived from the validation based on in-situ measurements is sufficient. We added error bars to the annual yield estimates of Figure 5 (former Figure 3) to highlight these.*

**3 The results suggest that all approaches are highly sensitive to mowing dates, with the estimated annual grassland yields showing a strong correlation to the number of mowing events. To assess the stability and effectiveness of each model, I recommend conducting a sensitivity test by removing the number of mowing events from all three models. This would allow for an evaluation of how each model performs with the remaining input variables**

*Thank you for highlighting this aspect and the suggestion to test this. We agree that the idea to test how the models perform without mowing data sounds interesting as this seems to be the most important input variable. However, all approaches rely on the mowing dates as input. There are no yield estimates without harvesting information. Therefore, we can't test this using these approaches and our model setups, unfortunately. However, as mentioned above added additional plots to the Appendix (Figure A3) showing the relationships of yields to temperature, precipitation and elevation per mowing frequency.*

**4 The annual grassland biomass estimation is based on a particular region in this study. It would be better to give some insights in the discussion section to show the possibility and potential use case to apply those three approaches to a bigger region, what the limitations (such as data availability) would be, and what method(s) will be most likely to serve better.**

*Thanks for this important comment. In the current version, we missed discussing the transferability of the approaches to other regions. We added this to the discussion in the revised manuscript (compare LL 614-621).*

Specific comments:

**1 Figure 3. Related to my comment #2, try to add an uncertainty range for the annual AGB**

*Thank you for this suggestion. We added the estimated errors to the annual yields of Figure 5 (former Figure 3) to show uncertainty ranges.*

**2 Figure 5. Try to reverse the color bar by using red as positive and blue as negative**
*Thanks for this comment. We followed your suggestion and changed the colors to make the plot more intuitive in Figure 7 (former Figure 5).*

**3 Line 430 - 440 & Figure 9. Is the correlation between annual yields and temperature, precipitation, and elevation obtained without dropping dominant mowing events? If so consider eliminating mowing events from the model and test the sensitivity**

*Thank you for your remark. The mowing events were not dropped here. But we tested the relationships per mowing frequency (Appendix Figure A3). The results stayed relatively the same.*